# Maximising the Utility of Validation Sets for Imbalanced Noisy-label Meta-learning

**Dung Anh Hoang**                                    *hoang.dung@monash.edu*
*Department of Data Science and AI*
*Monash University*

**Cuong Nguyen**                                      *c.nguyen@surrey.ac.uk*
*Centre for Vision, Speech and Signal Processing*
*University of Surrey*

**Vasileios Belagiannis**                             *vasileios.belagiannis@fau.de*
*Friedrich-Alexander-Universität Erlangen-Nürnberg*

**Thanh-Toan Do**                                     *toan.do@monash.edu*
*Department of Data Science and AI*
*Monash University*

**G. Carneiro**                                       *g.carneiro@surrey.ac.uk*
*Centre for Vision, Speech and Signal Processing*
*University of Surrey*

**Reviewed on OpenReview:** *https://openreview.net/forum?id=SBM9yeNZz5*

## Abstract

Meta-learning is an effective method to handle imbalanced and noisy-label learning, but it generally depends on a clean validation set. Unfortunately, this validation set has poor scalability when the number of classes increases, as traditionally these samples need to be randomly selected, manually labelled and balanced-distributed. This problem therefore has motivated the development of meta-learning methods to automatically select validation samples that are likely to have clean labels and balanced class distribution. Unfortunately, a common missing point of existing meta-learning methods for noisy label learning is the lack of consideration for data informativeness when constructing the validation set. The construction of an informative validation set requires hard samples, i.e., samples that the model has low confident prediction, but these samples are more likely to be noisy, which can degrade the meta reweighting process. Therefore, the balance between sample informativeness and cleanness is an important criteria for validation set optimization. In this paper, we propose new criteria to characterise the utility of such meta-learning validation sets, based on: 1) sample informativeness; 2) balanced class distribution; and 3) label cleanliness. We also introduce a new imbalanced noisy-label meta-learning (INOLML) algorithm that automatically builds a validation set by maximising such utility criteria. The proposed method shows state-of-the-art (SOTA) results compared to previous meta-learning and noisy-label learning approaches on several noisy-label learning benchmarks.

## 1 Introduction

The development of new deep learning methods will gradually depend more on poorly curated datasets (Xiao et al., 2015; Li et al., 2017). Such datasets tend to have class-imbalanced distributions and to contain large amounts of noisy-label samples. The problems of class-imbalanced learning and noisy-label learning have, however, been addressed separately. For instance, while noisy-label methods are based on robust

loss functions (Wang et al., 2023; 2019), label cleaning (Jaehwan et al., 2019; Yuan et al., 2018), meta-learning (Ren et al., 2018; Han et al., 2019), ensemble learning (Miao et al., 2015), and semi-supervised learning (Li et al., 2020), imbalanced learning approaches often rely on meta-learning (Ren et al., 2018; Han et al., 2019; Zhang & Pfister, 2021), transfer learning (Chu et al., 2020; Wang et al., 2017), classifier design (Wu et al., 2021; Liu et al., 2020), and re-sampling (Wang et al., 2020). Notice from the methods above that meta-learning methods have the potential to address both noisy-label and imbalanced learning problems (Ren et al., 2018; Zhang et al., 2020; Zhang & Pfister, 2021; Xu et al., 2021; Shu et al., 2019).

Meta-learning is often formulated as a bi-level optimization, where the upper-level optimization estimates the meta-parameters of interest using a validation set, while the lower-level optimization trains a model using a training set. In meta-learning, the validation set is commonly built by randomly selecting and manually labelling a class-balanced set of samples. However, the process of building these validation sets scales poorly when the number of classes escalates. Such issue, therefore, motivates the design of methods that automatically build validation sets containing pseudo-clean and class-balanced samples (Zhang & Pfister, 2021; Xu et al., 2021). However, the prediction accuracy results of such methods are lower than the accuracy from conventional approaches that rely on manually-curated validation sets (Zhang et al., 2020). We argue that this poor performance might be attributed to the low informativeness of the validation sets based on pseudo-clean samples (Zhang & Pfister, 2021; Xu et al., 2021), comparing to the potentially more informative randomly selected samples. The reason is that these methods tend to prioritize only high-confidence samples, as they are more likely to be clean, but such samples are often less informative to the model. As a result, defining *sample informativeness* within a meta reweighting framework emerges as a crucial challenge. At the same time, intuitively, *informative samples* are more likely to be *hard sample* (samples with low confident prediction, high gradient magnitude from the model), thus also more likely to be noisy, which can contaminate the validation set and degrade the performance of the meta learning framework. This issue has highlighted the need for a framework that can automatically construct and optimize the balance between informativeness and the integrity of the validation set.

Motivated by the bi-level optimization of the meta-learning algorithm, we propose novel criteria to characterise the utility of validation sets used in meta-learning. In particular, the proposed criteria are based on: 1) sample informativeness; 2) class-balanced distribution; and 3) label cleanliness. We also introduce a new imbalanced noisy-label meta-learning (INOLML) method that automatically builds a validation set by maximising its utility according to our proposed criteria. The proposed method, depicted in Figure 1, consists of 3 iterative steps: 1) detecting pseudo-clean samples from the noisy training set and labelling them; 2) forming the validation set from the robustly-labelled pseudo-clean set in step (1), using the proposed utility criteria; and 3) meta-learning using the validation set from step (2). The main contributions of our paper can be summarised as follows:

- A new criteria set to form the meta-learning validation set based on sample informativeness, class-balanced distribution, and label cleanliness.

- An innovative meta-learning algorithm (Figure 1), which automatically builds a validation set by maximising its utility according to our criteria, comprising the following steps: 1) detection and robust labelling of pseudo-clean samples from the noisy training set; 2) formation of the validation set using the proposed utility criteria; and 3) meta-learning using the validation set from step (2).

- A state-of-the-art meta-learning method to form validation sets for meta-learning that outperforms prior approaches that rely on random selecting or manually labelling.

In addition, the proposed meta-learning method INOLML shows competitive results with respect to previous meta-learning and noisy-label learning approaches on noisy-label learning benchmarks.

# 2 Related Work

## 2.1 Noisy-label learning

Noisy-label learning methods rely on many strategies, namely: robust loss functions (Charoenphakdee et al., 2019; Zhang & Sabuncu, 2018; Ghosh et al., 2017), label cleaning (Yuan et al., 2018; Jaehwan et al., 2019), co-teaching (Li et al., 2020; Jiang et al., 2018; Han et al., 2018), iterative label correction (Chen et al., 2021b; Arazo et al., 2019; Tu et al., 2023), semi-supervised learning (Ortego et al., 2021a; Li et al., 2020; Ortego et al., 2021b), contrastive learning (Huang et al., 2023), meta-learning (Ren et al., 2018; Zhang et al., 2020; Zhang & Pfister, 2021; Xu et al., 2021; Shu et al., 2019), and hybrid methods (Nguyen et al., 2020; Jiang et al., 2020). Except for the meta-learning approaches (Ren et al., 2018; Zhang et al., 2020; Zhang & Pfister, 2021; Xu et al., 2021; Shu et al., 2019), the majority of these methods assume a class-balanced training dataset.

For meta-learning approaches (Ren et al., 2018; Zhang et al., 2020; Zhang & Pfister, 2021; Xu et al., 2021; Shu et al., 2019), meta-parameters are introduced to automatically down-weight the losses of noisy samples and up-weight the losses of clean samples (Dehghani et al., 2017b;a). Such meta-parameters are then learnt by optimising the one-step-ahead model on a validation set.

To deal with noisy labels effectively, most of existing meta-learning approaches require a clean validation set. Such a set is, however, expensive to acquire and does not scale well with the number of classes. Recent studies relax such assumption by selecting the validation set from the noisy training set (Zhang & Pfister, 2021; Xu et al., 2021). Unfortunately, their sample selection process demonstrate lower performance compared to their counterparts using real validation data. This motivates us to develop an approach that can automatically build a clean and balanced validation set, but unlike them, our approach is motivated by the meta-learning optimization that also takes into consideration the informativeness of the validation samples.

## 2.2 Imbalanced learning

Imbalanced learning can result in a biased model with good accuracy for majority classes, but poor performance for the minority ones (Zhang et al., 2023). Methods to address imbalanced learning are based on transfer learning (Chu et al., 2020; Wang et al., 2017), classifier design (Liu et al., 2020; Wu et al., 2021), cost-sensitive learning (Zhou & Liu, 2006; Sun et al., 2007; Elkan, 2001), data augmentation (Zang et al., 2021; Chou et al., 2020), logit adjustment (Menon et al., 2021; Provost, 2008) and representation learning (Zhang et al., 2017; Huang et al., 2016). These existing methods often assume that training labels are clean, which does not hold for noisy-label datasets.

## 2.3 Noisy-label and imbalanced learning

Most of the methods mentioned in Sections 2.1 and 2.2 treat noisy-label and imbalance learning as two separate problems. Label noise in imbalanced datasets has also been considered by non meta-learning approaches (Cao et al., 2021; Wei et al., 2021; Karthik et al., 2021), but they either have different setups or achieve sub-par results. For meta-learning, the validation set is crucial to allow good model performance, where we empirically observe that different randomly selected clean validation sets can lead to substantially different performances. For example, if the validation set only contains "easy samples" that lie far from classification boundaries, meta-learning tends to produce poor classification accuracy. Unfortunately, previous approaches have not studied this issue. For instance, classic meta-learning approaches (Ren et al., 2018; Shu et al., 2019) rely on random sample selection and manual labelling, potentially producing uninformative validation sets. Furthermore, the fact that such manually-curated validation set is fixed for the whole training process may hinder model generalisation because the meta-learning optimization can quickly overfit to this validation set. Recent meta-learning approaches try to automatically build clean validation sets with "easy samples", obtained from low-loss samples (Xu et al., 2021) or well-optimised samples (Zhang & Pfister, 2021). Instead of selecting random samples or "easy samples", we study how to select informative samples for meta-learning.

# 3 Background

This section presents an overview of the label noise problem and two meta-learning methods widely-used when dealing with noisy-label datasets.

## 3.1 Noisy-label learning

In the conventional supervised learning, we are given a training set $\mathcal{D}_{\text{clean}} = \{(\mathbf{x}_i, \mathbf{t}_i)\}_{i=1}^{|\mathcal{D}_{\text{clean}}|}$, where $\mathbf{x}_i \in \mathcal{X} \subseteq \mathbb{R}^D$ represents a $D$-dimensional input data, and $\mathbf{t}_i \in \mathcal{Y} = \{\mathbf{t} : \mathbf{t} \in \{0,1\}^C \wedge \mathbf{1}^\top \mathbf{t} = 1\}$ is a $C$-dimensional one-hot clean label. The aim is to find a classification model $f_\theta : \mathcal{X} \to \Delta_{C-1}$, parameterised by $\theta \in \Theta$, that maps the input data to its corresponding label, where $\Delta_{C-1} = \{\mathbf{p} : \mathbf{p} \in [0,1]^C \wedge \mathbf{1}^\top \mathbf{p} = 1\}$ denotes the $(C-1)$-dimensional probability simplex.

In practice, instead of observing the correctly-labelled training set $\mathcal{D}_{\text{clean}}$, we are given a noisy training set $\mathcal{D} = \{(\mathbf{x}_i, \mathbf{y}_i)\}_{i=1}^{|\mathcal{D}|}$, where the noisy label $\mathbf{y}_i \in \mathcal{Y}$ might or might not be the same as the clean label $\mathbf{t}_i$. The aim is to exploit the information in such noisily-annotated training set to still learn a good model $f_\theta$ that can accurately predict the clean label $\mathbf{t}$ for an instance $\mathbf{x}$.

## 3.2 Learning to reweight

*Learning to reweight* (Ren et al., 2018) is a meta-learning approach proposed to address the label noise problem defined in Section 3.1. Given the nature of meta-learning, the original training set is split into two non-overlapping subsets: training (or support) set $\mathcal{D}^{(t)}$ and validation (or query) set $\mathcal{D}^{(v)}$. The main idea is to employ the validation subset $\mathcal{D}^{(v)}$ to learn a meta-parameter $\omega \in \Delta_{|\mathcal{D}^{(t)}|-1}$ (in the form of a probability vector) that weights the cross-entropy loss $\ell$ of each training sample in the training subset $\mathcal{D}^{(t)}$. The process obtaining $\omega$ is based on maximising some utility criteria, e.g., informativeness, class-balanced distribution and label cleanliness (Ren et al., 2018). Formally, the objective function of this meta-learning approach is a bi-level optimization defined as:

$$\min_\omega \mathbb{E}_{(\mathbf{x}_j, \mathbf{y}_j) \in \mathcal{D}^{(v)}} \left[ \ell\left(f_{\theta^*(\omega)}(\mathbf{x}_j), \mathbf{y}_j\right) \right] \quad \text{s.t.:} \ \theta^*(\omega) = \arg\min_\theta \mathbb{E}_{(\mathbf{x}_i, \mathbf{y}_i) \in \mathcal{D}^{(t)}} \left[ \omega_i \ell\left(f_\theta(\mathbf{x}_i), \mathbf{y}_i\right) \right], \tag{1}$$

where the notation $\mathbb{E}$ denotes the expectation and can be expressed as $\mathbb{E}_{\mathbf{u} \in \mathcal{U}}[g(\mathbf{u})] = {}^1\!/|\mathcal{U}| \sum_{\mathbf{u}_i \in \mathcal{U}} g(\mathbf{u}_i)$.

Intuitively, the lower-level (or the constraint) in (1) learns a classifier with weighted cross-entropy loss on the training subset $\mathcal{D}^{(t)}$, while the upper-level evaluate the performance of the classifier on the training validation subset $\mathcal{D}^{(v)}$ and optimises that performance w.r.t. the meta-parameter $\omega$.

The bi-level optimization in (1) can be solved by iterating the following two steps. In the first step, the parameter of the meta-learning model, $\theta^*(\omega)$, is estimated from the stochastic gradient descent (SGD) on the training subset $\mathcal{D}^{(t)}$, with each step defined by:

$$\theta^*(\omega) = \theta_0 - \eta_\theta \boldsymbol{\nabla}_\theta \mathbb{E}_{(\mathbf{x}_i, \mathbf{y}_i) \in \mathcal{D}^{(t)}} \omega_i \, \ell\left[(f_\theta(\mathbf{x}_i), \mathbf{y}_i)\right], \tag{2}$$

where $\eta_\theta$ is the step size or learning rate.

In the second step, the meta-parameter $\omega$ in the upper-level is updated by applying one SGD step on the loss evaluated on the validation subset $\mathcal{D}^{(v)}$ using the classifier's parameter $\theta^*(\omega)$ obtained in (2). The update is defined as:

$$\omega^* = \omega_0 - \eta_\omega \boldsymbol{\nabla}_\omega \mathbb{E}_{(\mathbf{x}_j, \mathbf{y}_j) \in \mathcal{D}^{(v)}} \left[ \ell\left(f_{\theta^*(\omega)}(\mathbf{x}_j), \mathbf{y}_j\right) \right], \tag{3}$$

where $\eta_\omega$ is the step size to update $\omega$.

The meta-parameter $\omega^*$ obtained in Eq. (3) is projected to the $(|\mathcal{D}^{(t)}|-1)$-dimensional probability simplex, $\Delta_{|\mathcal{D}^{(t)}|-1}$, before being used to train the classifier of interest with the weighted loss defined in the lower-level of (1).

### 3.3 Meta-relabelling

Based on the *learning to reweight* approach presented in Section 3.2, the *Distill* method (Zhang et al., 2020) proposes a slight modification of the objective function in (1) as follows:

$$\min_{\omega,\lambda} \mathbb{E}_{(\mathbf{x}_j,\mathbf{y}_j)\in\mathcal{D}^{(v)}} \left[\ell\left(f_{\theta^*(\omega,\lambda)}(\mathbf{x}_j),\mathbf{y}_j\right)\right] \quad \text{s.t.: } \theta^*(\omega,\lambda) = \underset{\theta}{\arg\min} \,\mathbb{E}_{(\mathbf{x}_i,\mathbf{y}_i)\in\mathcal{D}^{(t)}} \left[\omega_i\ell\left(f_\theta(\mathbf{x}_i),\hat{\mathbf{y}}(\lambda_i)\right)\right], \tag{4}$$

in which $\lambda = \{\lambda_i\}_{i=1}^{|\mathcal{D}^{(t)}|}$ is considered as another meta-parameter representing the tradeoff between the original label and model prediction. For each training sample $\mathbf{x}_i$ in the training subset $\mathcal{D}^{(t)}$, *Distill* uses two labels, $\hat{\mathbf{y}}_i(\lambda_0)$ for sample reweighting and $\mathbf{y}_i^*(\lambda_i^*)$ for sample relabeling. The modified label $\hat{\mathbf{y}}_i$ given the (noisy) label $\mathbf{y}_i$ and trade-off variable $\lambda_i$ is defined as:

$$\hat{\mathbf{y}}_i(\lambda_i) = \lambda_i\mathbf{y}_i + (1-\lambda_i)f_\theta(\mathbf{x}_i), \tag{5}$$

where $\lambda_i = 0.9 \quad \forall i = 1,2,\ldots,|\mathcal{D}^{(t)}|$. For label $\hat{\mathbf{y}}_i(\lambda_i)$, instead of optimizing the parameter $\lambda_0$, its value is fixed at 0.9 (close to 1) to ensure that label $\hat{\mathbf{y}}_i(\lambda_i)$ retains information from the original label. Supervised learning with label $\hat{\mathbf{y}}_i(\lambda_i)$ and the optimized weight $\omega_i$ effectively corresponds to upweighting clean samples and downweighting noisy samples.

On the other hand, the parameter $\lambda_i^*$ for the new label $\mathbf{y}_i^*(\lambda_i^*)$ is optimized with Eq.6 to infer the correct label for the samples, enabling uniform supervised learning. The pseudo-label $\mathbf{y}_i^*$ is defined as:

$$\mathbf{y}_i^* = \begin{cases} \mathbf{y}_i & \text{if } \lambda_i^* > 0 \\ f_\theta(\mathbf{x}_i) & \text{otherwise,} \end{cases}$$

where $\lambda_i^*$ is the gradient w.r.t. $\lambda_i$ of the final loss over the validation set $\mathcal{D}(v)$:

$$\lambda_i^* = \left[\mathsf{sign}\left(-\mathbb{E}_{(\mathbf{x}_j,\mathbf{y}_j)\in\mathcal{D}^{(v)}}\left[\frac{\partial}{\partial\lambda_i}\ell\left(f_{\theta^*(\omega,\lambda)}(\mathbf{x}_j),\mathbf{y}_j\right)\right]\right)\right]_+, \tag{6}$$

where $[\mathsf{sign}(.)]_+ = \max(\mathsf{sign}(.),0)$ denotes the rectification operator, and $\mathsf{sign}$ represents the sign function.

The estimated $\omega^*$, $\hat{\mathbf{y}}_i(\lambda_i)$ and $\mathbf{y}_i^*$ are then used to train a classifier of interest (Zhang et al., 2020). Given the pseudo label $\mathbf{y}_i^*$, the *Distill* method further employs supervised learning with images obtained via the *mixup* operator (Zhang et al., 2018) using the training and validation sets. Additionally, they adopt a KL-divergence loss between the model's outputs on the original and augmented inputs to enhance the consistency of the pseudo-label distribution. The final objective of *Distill* framework is defined as:

$$\mathsf{L} = \mathbb{E}_{(\mathbf{x}_i,\mathbf{y}_i)\in\mathcal{D}^{(t)}} \left[\omega_i^*\ell\left(f_\theta(\mathbf{x}_i),\hat{\mathbf{y}}_i(\lambda_i)\right) + \frac{\ell\left(f_\theta(\mathbf{x}_i),\mathbf{y}_i^*(\lambda_i^*)\right)}{B} + p\,\ell\left(\mathbf{y}_i^\beta, f_\theta(\mathbf{x}_i^\beta)\right) + k\,\mathrm{KL}\left[f_\theta\left(\mathbf{x}_i\right)\|f_\theta\left(\hat{\mathbf{x}}_i\right)\right]\right], \tag{7}$$

$\mathbf{y}_i^\beta$ and $\mathbf{x}_i^\beta$ are obtained via the *mixup* operator (Zhang et al., 2018) using the training and validation sets, $\mathrm{KL}[.\|.]$ denotes the Kullback-Leibler (KL) divergence, $\hat{\mathbf{x}}_i$ is an augmented sample of $\mathbf{x}_i$, and $p$ and $k$ are hyper-parameters, $B$ is the batch size.

## 4 Methodology

The meta-learning Distill model (Zhang et al., 2020) presented in Section 3.3 optimises meta-parameters $\lambda$ and $\omega$ under the guidance of a validation subset $\mathcal{D}^{(v)}$, and we adopt a similar approach for our framework. However, instead of selecting a fixed validation subset at the beginning of training as the Distill model, we propose a definition for *sample informativeness* within a meta reweighting framework, as well as an iterative mechanism to automatically select a "high-utility" validation subset at the start of each epoch. This section presents the utility criteria to select and the mechanism to update such a validation set.

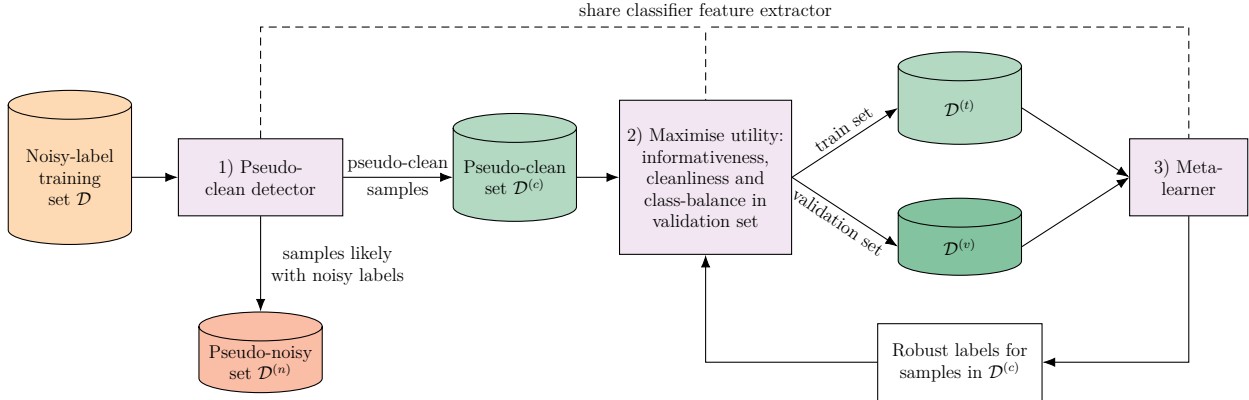

Figure 1: Main stages of INOLML: 1) classify the noisy-label samples from $\mathcal{D}$ into $\mathcal{D}^{(c)}$ (samples that are likely to have clean labels) and $\mathcal{D}^{(n)}$ (samples likely to have noisy labels); 2) build a validation set $\mathcal{D}^{(v)}$ containing samples that are informative (from a meta-learning perspective), balanced and with a high likelihood of containing clean labels, and 3) train the meta-learning classifier with $\mathcal{D}^{(t)} = \mathcal{D}^{(c)} \setminus \mathcal{D}^{(v)}$ and $\mathcal{D}^{(v)}$.

Specifically, we propose a 2-step process (corresponding to steps 1 and 2 in Figure 1 to select a high-utility validation set to train a meta-learning model presented in (4). In the first step, we split the original noisy training set $\mathcal{D}$ into pseudo-clean set $\mathcal{D}^{(c)}$ and pseudo-noisy set $\mathcal{D}^{(n)}$. In the second step, we further split the pseudo-clean set $\mathcal{D}^{(c)}$ into a training subset $\mathcal{D}^{(t)}$ and a validation subset $\mathcal{D}^{(c)}$ based on some criteria, such as cleanliness, informativeness and class-balanced distribution.

## 4.1 Detecting noisy and clean label subsets from $\mathcal{D}$

Formally, the original training set is split into two subsets: clean and noisy, following a certain criterion:

$$\mathcal{D}^{(c)} = \mathsf{PseudoCleanDetector}\,(\mathcal{D}) \quad \wedge \quad \mathcal{D}^{(n)} = \mathcal{D} \setminus \mathcal{D}^{(c)}. \tag{8}$$

We first calculate the cross-entropy losses between noisy labels and the prediction $f_\theta(.)$ of all samples in the training set $\mathcal{D}$. We then apply the *small loss hypothesis* (Han et al., 2018; Li et al., 2020) through the notation $\mathsf{PseudoCleanDetector}(.)$ in (8) to select samples having small losses to form a pseudo-clean set $\mathcal{D}^{(c)}$, because they are more likely to be clean. Note that the pseudo-clean set $\mathcal{D}^{(c)}$ is independent from the target model since $\mathcal{D}^{(c)}$ is initialised using a warm-up model (i.e., a model trained on $\mathcal{D}$ for a few epochs), while the target model is trained from scratch. Hence, the small-loss samples in the pseudo-clean set $\mathcal{D}^{(c)}$ are likely informative to the target model. The remaining samples form the pseudo-noisy set $\mathcal{D}^{(n)}$, as shown in (8). These two sets, $\mathcal{D}^{(c)}$ and $\mathcal{D}^{(n)}$, are regularly updated during training.

## 4.2 Maximising the utility of the validation set

The pseudo-clean set $\mathcal{D}^{(c)}$ obtained in (8) is then divided into a validation set $\mathcal{D}^{(v)}$ and a training set $\mathcal{D}^{(t)}$. The validation set $\mathcal{D}^{(v)}$ must have the following properties:

- *class-balanced:* containing the same number of samples per class.

- *informative:* being useful to guide the meta-learning framework. We define validation samples with high *informativeness* as samples that can assign larger weight $\omega_i$ for clean samples, and lower weight for noisy samples.

- *clean:* most likely having clean labels.

To enforce such properties, we propose the objective function of the validation set as follows:

$$\mathcal{D}^{(t)} = \mathcal{D}^{(c)} \setminus \mathcal{D}^{(v)}_{\text{opt}}$$

$$\mathcal{D}^{(v)}_{\text{opt}} = \underset{\substack{\mathcal{D}^{(v)} \subset \widetilde{\mathcal{D}}^{(v)}_{\text{opt}} \\ \left|\mathcal{D}^{(v)}\right| = M \times C}}{\text{argmax}} \ \text{Clean}\left(\mathcal{D}^{(v)}, \mathcal{D}^{(c)}\right)$$

$$\text{s.t.:} \ \widetilde{\mathcal{D}}^{(v)}_{\text{opt}} = \underset{\substack{\widetilde{\mathcal{D}}^{(v)} \subset \mathcal{D}^{(c)} \\ \left|\widetilde{\mathcal{D}}^{(v)}\right| = K \times C}}{\text{argmax}} \ \text{Info}\left(\widetilde{\mathcal{D}}^{(v)}, \mathcal{D}^{(c)}\right), \tag{9}$$

where $K$ and $M$ are the number of samples per class with $M \leq K$ to guarantee a class-balanced distribution for $\mathcal{D}^{(v)}$, and the functions $\text{Clean}()$ and $\text{Info}()$ denote the cleanliness and informativeness of samples. $\widetilde{\mathcal{D}}^{(v)}_{\text{opt}}$ and $\mathcal{D}^{(v)}_{\text{opt}}$ respectively denote the optimized $\widetilde{\mathcal{D}}^{(v)}$ and $\mathcal{D}^{(v)}$.

In other words, the lower-level in (9) means to determine a "coarse" validation set $\widetilde{\mathcal{D}}^{(v)}$ that is class-balanced (containing $K$ samples per class) and informative from the pseudo-clean set $\mathcal{D}^{(c)}$. The upper-level in (9) refines the coarse validation set $\widetilde{\mathcal{D}}^{(v)}$ further to form a "fine" subset $\mathcal{D}^{(v)}$ containing most likely clean samples. The need for this bi-level optimization is because in practice, as the training progress, we find that informative samples need to become harder samples (samples with low confident prediction from the model) over time, hence more likely to be noisy. Note that while clean and informative samples are beneficial for the meta reweighting framework, informative-but-noisy validation samples will make the model prone to overfitting. Our optimization for the validation set $\mathcal{D}^{(v)}$ hence aim to find the balance between class-balance, informativeness and cleanness. The details of $\text{Clean}(.)$ and $\text{Info}(.)$, are presented in the following subsubsections.

### 4.2.1 Informativeness

As described above, *informative* samples for the validation set of a meta reweighting framework should be able to upweight clean samples, and downweight noisy samples effectively. Based on this motivation, our $\text{Info}(.)$ function in the lower-level of (9) is motivated by *learning to reweight* (Ren et al., 2018) (similar to (1)). According to (Ren et al., 2018, Eq. (12)), the gradient w.r.t. $\omega$ can be written as follows:

$$\mathbb{E}_{(\mathbf{x}_j, \mathbf{y}_j) \in \mathcal{D}^{(v)}} \left[ \frac{\partial}{\partial \omega_i} \ell\left(f_{\theta^*(\omega, \lambda)}(\mathbf{x}_j), \mathbf{y}_j\right) \bigg|_{\omega_i = 0} \right] \propto -\mathbb{E}_{(\mathbf{x}_j, \mathbf{y}_j) \in \mathcal{D}^{(v)}} \sum_{l=1}^{L} \left( \mathbf{z}^{(v)\top}_{j,l-1} \mathbf{z}^{(t)}_{i,l-1} \right) \left( \mathbf{g}^{(v)\top}_{j,l} \mathbf{g}^{(t)}_{i,l} \right), \tag{10}$$

where $\mathbf{z}^{(v)}_{j,l-1}$ is the feature of the validation sample $\mathbf{x}_j$ processed by a deep model at the layer $l$-th (similar definition is applied for the training feature $\mathbf{z}^{(t)}_{i,l-1}$), and $\mathbf{g}^{(v)}_{j,l}$ is the gradient from layer $l$ for the validation sample $\mathbf{x}_j$ (similar definition is applied for the training gradient $\mathbf{g}^{(t)}_{i,l}$).

The re-weighting factor $\omega_i$ of a training sample $\mathbf{x}_i$ is, therefore, high if its feature and gradient are similar to the validation samples' feature and gradient; else, the weight is low. Hence, a validation set that maximises the weight $\omega$ of training samples also maximises its meta-learning optimization utility. This observation is crucial to our validation sample selection. Intuitively, we can define the sample informativeness function $\text{Info}(.)$ that we want to maximize as below:

$$\text{Info}(\widetilde{\mathcal{D}}^{(v)}, \mathcal{D}^{(c)}) = \sum_{\substack{(\mathbf{x}_i, \mathbf{y}_i) \in (\mathcal{D}^{(c)} \setminus \widetilde{\mathcal{D}}^{(v)})}} \sum_{\substack{(\mathbf{x}_j, \mathbf{y}_j) \in \widetilde{\mathcal{D}}^{(v)} \\ \mathbf{y}_j = \mathbf{y}_i}} h(\mathbf{x}_i, \mathbf{x}_j), \tag{11}$$

where $h$ is analogous to a measure of similarity between samples:

$$h(\mathbf{x}_i, \mathbf{x}_j) = \sum_{l=1}^{L} (\mathbf{z}^\top_{j,l-1} \mathbf{z}_{i,l-1})(\mathbf{g}^\top_{j,l} \mathbf{g}_{i,l}), \tag{12}$$

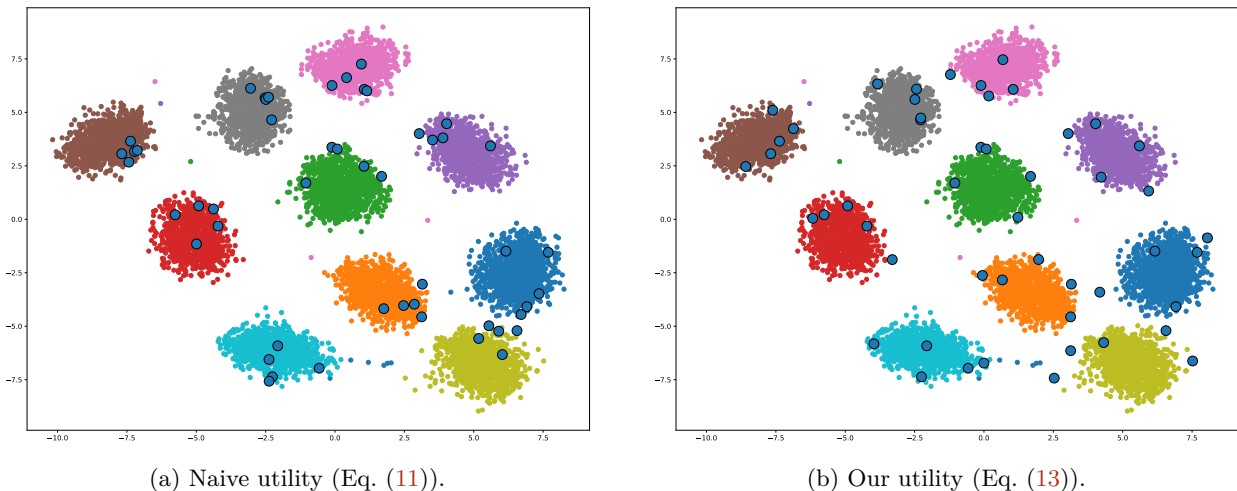

(a) Naive utility (Eq. (11)).  (b) Our utility (Eq. (13)).

Figure 2: Comparison between 2-dimensional t-SNE representations of the samples selected (samples for each class have different colours, and the selected validation samples per class are highlighted with a blue dot with black outline) by (a) naive utility in Eq. (11), and (b) our utility in Eq. (13). Note that this is the t-SNE representation for CIFAR10 dataset with a uniform noise rate of 40%.

with $\mathbf{z}_{j,l-1}$ being the feature of $\mathbf{x}_j$ at the input of layer $l$ (same for feature $\mathbf{z}_{i,l-1}$ of sample $\mathbf{x}_i$), and $\mathbf{g}_{j,l}$ being the validation gradient of layer $l$ from $\mathbf{x}_j$ (same for $\mathbf{g}_{i,l}$ from $\mathbf{x}_i$). Note that $h(.,.)$ in (12) is a part of the weight $\omega_i$ for the training sample $(\mathbf{x}_i, \mathbf{y}_i)$ (see (10)), and $h(x_i, x_j)$ represents how much the validation sample $(\mathbf{x}_j, \mathbf{y}_j)$ can contribute to the weight of the clean training sample $(\mathbf{x}_i, \mathbf{y}_i)$, consists of a gradient matching term $\mathbf{g}_{j,l}^\top \mathbf{g}_{i,l}$ and a feature matching term $\mathbf{z}_{j,l-1}^\top \mathbf{z}_{i,l-1}$ between the two samples. Ideally, we want the weight of the clean samples remain high throughout the training process. Because $h(x_i, x_j)$ depends on samples gradient $\{\mathbf{g}_{i,l}\}_{l=1}^L$, $\{\mathbf{g}_{i,l}\}_{l=1}^L$ that may vary greatly during the training process, maximizing the function from Eq. (11) may lead to a biased validation set that is optimal only for the current iteration, not throughout the training process. If a validation sample $(\mathbf{x}_j, \mathbf{y}_j)$ has high contribution to the weight of a clean training sample $(\mathbf{x}_i, \mathbf{y}_i)$, then they may share similar characteristics. Consequently, their gradient matching $\mathbf{g}_{j,l}^\top \mathbf{g}_{i,l}$ and feature matching $\mathbf{z}_{j,l-1}^\top \mathbf{z}_{i,l-1}$ will remain high and robust during the training process. Therefore, we need to modify the function $\mathsf{Info}(.)$, so that for each clean training sample $(\mathbf{x}_i, \mathbf{y}_i)$, there must be at least one validation sample $(\mathbf{x}_j, \mathbf{y}_j)$ that has high weight contribution to it. Such observation allows us to re-define the function $\mathsf{Info}(.)$ in the lower-level of (9) as follows:

$$\mathsf{Info}(\widetilde{\mathcal{D}}^{(v)}, \mathcal{D}^{(c)}) = \sum_{(\mathbf{x}_i, \mathbf{y}_i) \in (\mathcal{D}^{(c)} \setminus \widetilde{\mathcal{D}}^{(v)})} \max_{\substack{(\mathbf{x}_j, \mathbf{y}_j) \in \widetilde{\mathcal{D}}^{(v)} \\ \mathbf{y}_j = \mathbf{y}_i}} h(\mathbf{x}_i, \mathbf{x}_j), \tag{13}$$

In order to observe the effectiveness of this approach, we compare our modified $\mathsf{Info}(.)$ in Eq. (13) with the naive utility from Eq. (11). Figure 2 shows a visual comparison between the naive utility in Eq. (11) and our utility in Eq. (13) for the selection of validation samples from the pseudo-clean set $\mathcal{D}^{(c)}$ using t-SNE projection to a 2-dimensional space for CIFAR10 dataset with a uniform noise rate of 40%. It is noticeable that the selected validation samples using our proposed utility are diverse and distributed around the clusters of their respective classes, while the validation samples selected with the naive utility in Eq. (11) tend to be located closer to each other and inside each class cluster. This is because the naive utility favors low-confidence samples at the selection time. However, as training progresses and the prediction confidence of these samples increases, other samples should become the target for upweighting instead, making the selected validation set no longer optimal. Another observation is that our utility maximization results in a validation set that constructs a boundary between each class in feature space, helping our method to identify the classification boundary between different classes.

---

**Algorithm 1** Training procedure of the proposed INOLML.

---

1: **procedure** TRAIN($\mathcal{D}$, $\eta$, $T$, $\widetilde{T}$, $T^{(u)}$, $\widetilde{\eta}$, $\kappa$, $N$, $M$, $K$, $C$)
2:     ▷ *$\mathcal{D}$: noisy training set, $\{\eta_t\}_t^T$: learning rates*             ◁
3:     ▷ *$T$: total number of iterations*             ◁
4:     ▷ *$\widetilde{T}$: minimum number of iterations before updating the robust labels*             ◁
5:     ▷ *$T^{(u)}$: interval between updates*             ◁
6:     ▷ *$\kappa, N, M, K, C$: hyper-parameters*             ◁
7:     Warm-up $f_\theta(.)$ with $\ell_{\text{CE}}(.)$ on $\mathcal{D}$
8:     $\mathcal{D}^{(c)} \leftarrow$ PseudoCleanDetector $(\mathcal{D})$          ▷ *Eq.* (8)
9:     Initialise robust label $\{\tilde{\mathbf{y}}_i\}_{i=1}^{|\mathcal{D}^{(c)}|}$ of samples in $\mathcal{D}^{(c)}$
10:     Initialise $\mathcal{D}^{(v)}$ and $\mathcal{D}^{(t)}$ from $\mathcal{D}^{(c)}$          ▷ *Eq.* (9)
11:     Reinitialize $f_\theta(.)$
12:     **for** $t = 1$ to $T$ **do**
13:        Meta-learn $\theta$, $\omega$ and $\lambda$          ▷ *Eq.* (4)
14:        **if** $(t > \widetilde{T})$ **then**
15:           Update $\{\tilde{\mathbf{y}}_i\}_{i=1}^{|\mathcal{D}^{(c)}|}$ of samples in $\mathcal{D}^{(c)}$          ▷ *Eq.* (15)
16:        **if** $t \mod T^{(u)} = 0$ **then**
17:           Construct $\widehat{\mathcal{D}}^{(c)}$.          ▷ *Eq.* (16)
18:           Select new $\mathcal{D}^{(v)}$ from $\widehat{\mathcal{D}}^{(c)}$          ▷ *Eq.* (9)
19:     **return** the trained model parameter $\theta$

---

### 4.2.2   Cleanliness

Note that our focus on the informativeness of the validation set should not compromise the cleanliness of samples, as meta-learning under the guidance of informative-but-noisy validation samples will make the model prone to overfitting. Indeed, the samples in the "coarse"-but-informative validation set $\widehat{\mathcal{D}}^{(v)}$ can still have noisy labels since $\mathcal{D}^{(c)}$ is not completely clean. In addition, the function $\mathsf{Info}(x_i, x_j)$ prioritizes samples in $\widetilde{\mathcal{D}}^{(v)}$ with large gradient magnitude, as it has a gradient matching component $\mathbf{g}_{j,l}^\top \mathbf{g}_{i,l}$. Consequently, the selected validation samples are usually "hard samples" and more likely to be noisy. It is, therefore, important to balance the trade-off between sample informativeness and cleanness. However, as gradient magnitude is one of our main factors to identify informative samples, we do not want to minimize their gradient magnitude for cleanliness metric. By observing that the samples from $\tilde{\mathcal{D}}^{(v)}$ are more likely to be clean when they have high similarity with other samples in the same class, we propose a heuristics to identify the samples of interest with high chance being clean. Specifically, we employ the cosine similarity between the samples of interest and other samples in the same class in $\mathcal{D}^{(c)}$, defined as follows:

$$\mathsf{Clean}\left(\mathcal{D}^{(v)}, \mathcal{D}^{(c)}\right) = \sum_{\substack{(\mathbf{x}_j, \mathbf{y}_j) \in \mathcal{D}^{(v)} \\ (\mathbf{x}_i, \mathbf{y}_i) \in \mathcal{D}^{(c)} \setminus \mathcal{D}^{(v)} \\ \mathbf{y}_i = \mathbf{y}_j}} \sum_{l=1}^{L} \left(\mathbf{z}_{j,l-1}^\top \mathbf{z}_{i,l-1}\right). \tag{14}$$

Both combinatorial optimizations in (9), and in particular Eqs. (13) and (14), are solved with greedy strategies. We initially loop through each class and select $K$ samples per class to maximise the lower-level objective function. We then sequentially select $M$ samples among the previous set of $K$ samples per each class to optimise the upper-level objective function. We simplify the calculation of gradient in (10) and the optimization in (9) by using only the features and gradients in the last layer of the deep neural network of interest. This simplification is reasonable since according to (Zhang & Pfister, 2021), the weights of training samples in meta-learning depend mostly on the last layer of the model, and for datasets with a large number of classes, the memory complexity is intractable. For instance, we need around $K = 200$ candidate samples per class for $\tilde{\mathcal{D}}^{(v)}$ before selecting $M = 10$ samples per class for the validation set $\mathcal{D}^{(v)}$ (e.g., for CIFAR100, we need 20,000 samples for $\tilde{\mathcal{D}}^{(v)}$), where for each sample, we have to store its gradient and feature embedding for every layer. For example, for Resnet18 that has $1 \times 10^7$ parameters, if we needed to store all features and

gradients from all layers for the whole $\tilde{\mathcal{D}}^{(v)}$, we would need to store $(2 \times 10^4) \times (1.1 \times 10^7)$ floating point numbers (i.e., $\approx 10^{12}$ bytes), which is intractable.

### 4.3 Dynamic Pseudo Clean Set Refinement

To further refine the pseudo clean set $\mathcal{D}^{(c)}$, we adopt the robust moving average label of each sample which is defined as follows:

$$\tilde{\mathbf{y}}_i = \kappa \tilde{\mathbf{y}}_i + (1 - \kappa)\frac{1}{E} \sum_{e=1}^{E} f_\theta(\mathbf{x}_i), \tag{15}$$

with $\kappa = 0.9$. The robust label is updated regularly, and becomes more accurate as the model improves. At the start of each data selection step, we eliminate potentially noisy samples from the pseudo clean set that were previously overlooked by the PseudoCleanDetector using the following method:

$$\widehat{\mathcal{D}}^{(c)} = \left\{ (\mathbf{x}_i, \mathbf{y}_i) : (\mathbf{x}_i, \mathbf{y}_i) \in \mathcal{D}^{(c)} \wedge \operatorname*{argmax}_{k \in \{1,...,C\}} \mathbf{y}_i(k) = \operatorname*{argmax}_{k \in \{1,...,C\}} \tilde{\mathbf{y}}_i(k) \right\}. \tag{16}$$

We utilize the proxy set $\widehat{\mathcal{D}}^{(c)}$ for our data selection algorithm instead of the full set $\mathcal{D}^{(c)}$.

### 4.4 Training procedure

Our training follows the 3-step iterative approach in Figure 1, where step 1 (pseudo-clean label detector) and step 2 (utility maximisation of the validation set) have been explained above, and step 3 (meta-learning) is based on (4) with the loss L defined in Eq. (7). The details of the training process are shown in Algorithm 1.

## 5 Experiments

### 5.1 Datasets

The proposed method INOLML is evaluated on several datasets, including CIFAR10, CIFAR100, mini-WebVision and Controlled Noisy Web Labels (CNWL). Both CIFAR10 and CIFAR100 datasets (Krizhevsky & Hinton, 2009) contain 50k and 10k images used for training and testing, respectively. Each image has a size of 32×32 pixels and is labelled as one of 10 or 100 classes. WebVision (Li et al., 2017) is a dataset of 2.4 million images crawled from Google and Flickr based on the 1,000 ImageNet classes (Deng et al., 2009). The dataset is more challenging than CIFAR since it is class-imbalanced and contains real-world noisy labels. Following (Zhang & Pfister, 2021), we extract a subset that contains the first 50 classes to create the mini-WebVision dataset (Jiang et al., 2018). CNWL (Jiang et al., 2020) is a benchmark of controlled real-world label noise that contains noise rates from 0 to 0.8. Following recent studies (Xu et al., 2021), we evaluate the proposed method on the Red mini-ImageNet dataset consisting of 50k training images from 100 classes for training and 5k images for testing. Note that we use the image size of 32×32 pixels for a fair comparison with FAMUS (Xu et al., 2021) and other related methods (Xu et al., 2021).

### 5.2 Implementation details

For all experiments on CIFAR datasets, except long-tail imbalance, we use the same hyperparameters and network architectures as the Distill model (Zhang et al., 2020). We adopt the cosine learning rate decay with warm restarting (Loshchilov & Hutter, 2017) and SGD optimiser. For CIFAR, we train WideResnet28-10 with 100k iterations and a batch size of 100. We also train a smaller network (Resnet29) to fairly compare with (Zhang et al., 2020). For mini-WebVision, we follow FSR (Zhang & Pfister, 2021) and train a single Resnet50 network with 1 million iterations and a batch size of 16. For Red mini-ImageNet, we run experiments with 150k iterations and a batch size of 100. For CNWL, we use a single PreAct Resnet18 network to be consistent with previous works (Cordeiro et al., 2021; Ortego et al., 2021b) on this benchmark. For the class imbalance problems, we use Resnet32 to fairly compare with FaMUS (Xu et al., 2021) and FSR (Zhang & Pfister, 2021). We report the prediction accuracy of each experiment on their corresponding

Table 1: Test accuracy (%) of INOLML and previous methods for symmetric noise; methods with $^{\text{T}}$ represent meta-learning methods that need clean validation sets; the lower block contains meta-learning methods, while the upper block shows SOTA methods.

| Method | CIFAR10 | | | CIFAR100 | | |
|---|---|---|---|---|---|---|
| | 0.2 | 0.4 | 0.8 | 0.2 | 0.4 | 0.8 |
| GJS | 95.3 ± 0.2 | 93.6 ± 0.2 | 79.1 ± 0.3 | 78.1 ± 0.3 | 75.7 ± 0.3 | 44.5 ± 0.5 |
| DivideMix | 95.7 ± 0.0 | - | 92.9 ± 0.0 | 76.9 ± 0.0 | - | 59.6 ± 0.0 |
| CRUST | 91.1 ± 0.2 | 89.2 ± 0.2 | 58.3 ± 1.8 | - | - | - |
| PENCIL | - | - | - | 73.9 ± 0.3 | 69.1 ± 0.6 | - |
| UNICON | 96.0 ± 0.0 | - | 93.9 ± 0.0 | 78.9 ± 0.0 | - | 63.9 ± 0.0 |
| CausalNL + NPC | 81.2 ± 0.0 | - | 18.8 ± 0.0 | - | - | - |
| DMLP$^{\text{T}}$ | 94.2 ± 0.0 | - | 93.2 ± 0.0 | 72.3 ± 0.0 | - | 63.2 ± 0.0 |
| DMLP-DivideMix$^{\text{T}}$ | 96.2 ± 0.0 | - | 94.3 ± 0.0 | 79.4 ± 0.0 | - | 68.5 ± 0.0 |
| TLC | 95.0 ± 0.1 | - | 92.5 ± 0.2 | 78.0 ± 0.2 | - | 65.0 ± 0.3 |
| PSDC | 96.2 ± 0.0 | - | 94.0 ± 0.0 | 79.4 ± 0.0 | - | 64.3 ± 0.0 |
| Bayesian DivideMix++ | 96.13 ± 0.07 | - | 94.97 ± 0.02 | 80.02 ± 0.03 | - | 70.01 ± 0.23 |
| Distill$^{\text{T}}$ | 96.2 ± 0.2 | 95.9 ± 0.2 | 93.7 ± 0.5 | 81.2 ± 0.7 | 80.2 ± 0.3 | **75.5 ± 0.2** |
| MentorNet$^{\text{T}}$ | 92.0 ± 0.0 | 89.0 ± 0.0 | 49.0 ± 0.0 | 73.0 ± 0.0 | 68.0 ± 0.0 | 35.0 ± 0.0 |
| L2R$^{\text{T}}$ | 90.0 ± 0.4 | 86.9 ± 0.2 | 73.0 ± 0.8 | 67.1 ± 0.1 | 61.3 ± 2.0 | 35.1 ± 1.2 |
| MWN$^{\text{T}}$ | 90.3 ± 0.6 | 87.5 ± 0.2 | - | 64.2 ± 0.3 | 58.6 ± 0.5 | - |
| GDW$^{\text{T}}$ | - | 88.1 ± 0.4 | - | - | 59.8 ±1.6 | - |
| FaMUS | - | 95.3 ± 0.2 | - | - | 76.0 ± 0.2 | - |
| FSR | 95.1 ± 0.1 | 93.7 ± 0.1 | 82.8 ± 0.3 | 78.7 ± 0.2 | 74.2 ± 0.4 | 46.7 ± 0.8 |
| **INOLML** | **96.9 ± 0.1** | **96.6 ± 0.1** | **95.0 ± 0.2** | **82.0 ± 0.2** | **81.3 ± 0.2** | 74.7±0.1 |

testing sets. We compare our method with recently published state-of-the-art (SOTA) meta-learning approaches, including FaMUS (Xu et al., 2021), FSR (Zhang & Pfister, 2021), Meta Weight Net (Shu et al., 2019), Distill (Zhang et al., 2020), GDW (Chen et al., 2021a), L2R(Ren et al., 2018),MSLG(Algan & Ulusoy, 2021). Furthermore, some SOTA noisy labels learning approaches are also being compared with, such as DivideMix (Li et al., 2020), CausalNL (Yao et al., 2021), NPC (Bae et al., 2022), MentorMix (Jiang et al., 2020), MentorNet (Jiang et al., 2018), MOIT (Ortego et al., 2021b), GJS (Englesson & Azizpour, 2021), CRUST (Mirzasoleiman et al., 2020), Co-teaching (Han et al., 2018), Iterative-CV (Chen et al., 2019), HAR (Cao et al., 2021), UNICON (Karim et al., 2022), NCR (Iscen et al., 2022), C2D (Zheltonozhskii et al., 2022), BtR (Smart & Carneiro, 2022), CC (Zhao et al., 2022), SSR (Feng et al., 2021), DMLP (Tu et al., 2023), Twin Contrast (Huang et al., 2023). For imbalance learning mixed with noisy label experiments, we compare with recent noisy-label imbalanced learning methods, including ROLT (Wei et al., 2021), FSR (Zhang & Pfister, 2021), LDAM (Cao et al., 2019), BBN (Zhou et al., 2020), HAR (Cao et al., 2021), and CRUST (Mirzasoleiman et al., 2020). The supplementary material shows more implementation details.

### 5.3 Symmetric noise

Table 1 shows the test accuracy results on symmetric noise rates varying from 20% to 80% for different meta-learning approaches that require a clean validation set (indicated with $^{\text{T}}$) and others that automatically build validation sets. INOLML outperforms all previous methods in most cases. The slightly lower performance than Distill on CIFAR100 at 80% noise rate can be explained by Distill's large manually-curated clean validation set with 10 clean samples per class. In addition, as shown in Figure 3a, at 80% symmetric noise rate, a significant proportion (20% to 45%) of our clean validation set $\mathcal{D}^{(v)}$ contains noisy samples at the final training stages. This, however, deteriorates the efficacy of our approach. Note that it is reasonable that the noise rate increases in the validation set as training progresses because our method prioritises the more informative samples (or harder samples) over clean ones to be included in the validation set at each

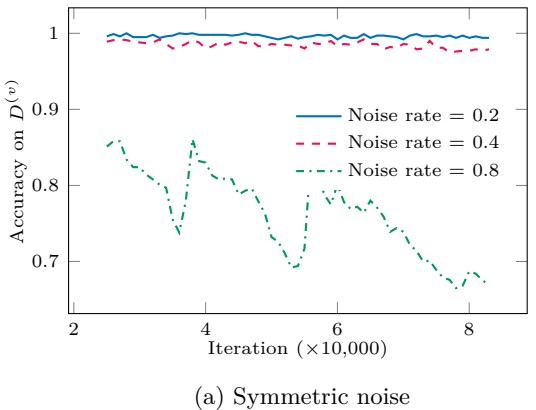 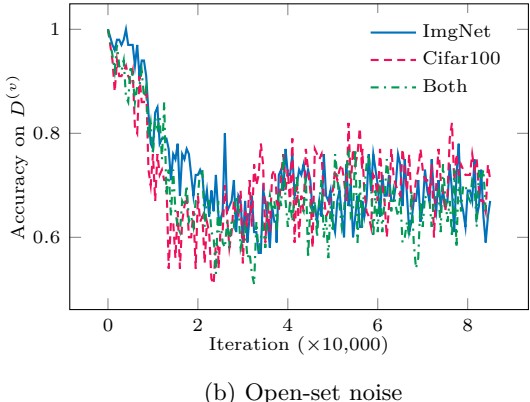

(a) Symmetric noise (b) Open-set noise

Figure 3: Accuracy of the clean validation set $\mathcal{D}^{(v)}$ as training progresses evaluated on different noise benchmarks.

Table 2: Test accuracy (%) of INOLML on CIFAR10 with 0.4 asymmetric noise, in comparison with Distill using a validation set $\mathcal{D}^{(v)}$ of sizes 1, 5 and 10 samples per class on Resnet29 and WideResnet28-10. The superscript $^\text{T}$ indicates the need for clean validation sets.

| Method | Size of validation set $\left|\mathcal{D}^{(v)}\right|$ | Resnet29 | WRN28-10 |
|---|---|---|---|
| Distill$^\text{T}$ | $1 \times C$ | $76.8 \pm 2.9$ | $93.2 \pm 0.2$ |
| **INOLML** | | $86.8 \pm 0.9$ | $93.6 \pm 0.3$ |
| Distill$^\text{T}$ | $5 \times C$ | $86.7 \pm 0.5$ | $94.5 \pm 0.2$ |
| **INOLML** | | $89.3 \pm 0.2$ | $94.1 \pm 0.1$ |
| Distill$^\text{T}$ | $10 \times C$ | $88.6 \pm 0.7$ | $\mathbf{94.9 \pm 0.1}$ |
| **INOLML** | | $\mathbf{89.8 \pm 0.3}$ | $94.2 \pm 0.1$ |

epoch. Such prioritisation results in a decrease in the cleanliness of our validation set as the model gets more accurate. We also carry out additional experiments with different validation set sizes to fairly compare with Distill in Appendix C. In particular, our method outperforms Distill by 1% to 3% in most scenarios. Overall, these results show that a pseudo-clean, balanced, and informative validation set, can outperform a randomly-collected clean validation set in symmetric noise scenarios. Our results also set new SOTA results on the symmetric label noise benchmarks for methods without a clean validation set.

### 5.4 Asymmetric noise

Table 2 shows a comparison between INOLML and Distill (Zhang et al., 2020) with different validation set sizes: 1, 5 and 10 samples per class. Although our proposed method does not rely on a manually-labelled validation set, it performs better than Distill, especially with small network architectures (Resnet29) and small validation set sizes (1 sample per class). Our method has slightly lower accuracy than Distill with larger clean validation set sizes (at least 5 random clean samples per classes) on larger network architectures (WideResnet28). This might be caused by the confirmation bias of asymmetric noise in our pseudo-clean validation subset and the high capacity of larger models, e.g., WideResnet28-10, which are more prone to overfit label noise when trained on a small dataset, like CIFAR10. In addition, INOLML also shows better performance than SOTA methods, like FSR, DivideMix and UNICON (see Table 3).

Table 3: Test accuracy (%) of INOLML and previous methods on CIFAR10 with 0.4 asymmetric noise (meta-learning approaches are in the bottom part of the table). The superscript $^T$ indicates the need for clean validation sets.

| Method | Accuracy |
|---|---|
| GJS | $89.7 \pm 0.4$ |
| F-Correction | $83.6 \pm 0.3$ |
| UNICON | $94.1 \pm 0.0$ |
| PENCIL | $91.2 \pm 0.0$ |
| DivideMix | $92.1 \pm 0.0$ |
| CausalNL | $74.8 \pm 0.0$ |
| TLC | $92.6 \pm 0.0$ |
| TLC+ | $93.7 \pm 0.0$ |
| PSDC | $94.2 \pm 0.0$ |
| **DMLP$^T$** | **$95.0 \pm 0.0$** |
| L2R$^T$ | $90.8 \pm 0.3$ |
| FSR | $93.6 \pm 0.3$ |
| **INOLML** | **$94.2 \pm 0.1$** |

Table 4: Test accuracy (%) of INOLML and other SOTA meta-learning approaches evaluated on the CIFAR imbalanced learning (long-tailed) recognition task. The reported results are from (Zhang & Pfister, 2021) and (Xu et al., 2021).

| | CIFAR10 | | | CIFAR100 | | |
|---|---|---|---|---|---|---|
| Imbalance ratio | **200** | **50** | **10** | **200** | **50** | **10** |
| Softmax | 65.68 | 74.81 | 86.39 | 34.84 | 43.85 | 55.71 |
| CB-Focal | 65.29 | 76.71 | 86.66 | 32.62 | 44.32 | 55.78 |
| CB-Best | 68.89 | 79.27 | 87.49 | 36.23 | 45.32 | 57.99 |
| L2R | 66.51 | 78.93 | 85.19 | 33.38 | 44.44 | 53.73 |
| MWN | 68.91 | 80.06 | 87.84 | 37.91 | 46.74 | 58.46 |
| GDW | - | - | 86.8 | - | - | 56.8 |
| FaMUS | - | 83.32 | 87.90 | - | 49.93 | 59.03 |
| FSR-DF | 66.15 | 79.78 | 88.15 | 36.74 | 44.43 | 55.60 |
| FSR | 67.76 | 79.17 | 87.40 | 35.44 | 42.57 | 55.45 |
| **INOLML** | **74.91** | **84.43** | **90.81** | **41.52** | **51.35** | **62.07** |

## 5.5 Imbalanced learning

We evaluate INOLML on the imbalanced CIFAR datasets following the setting in (Zhang & Pfister, 2021). The prediction accuracy in Table 4 shows that INOLML considerably surpasses other meta-learning approaches.

## 5.6 Imbalanced noisy-label learning

We evaluate our proposed method in the setting that combines class imbalance and label noise from (Zhang & Pfister, 2021). We follow the same experimental configuration by adding 20% and 40% symmetric noise to the CIFAR10 imbalanced datasets with different imbalance ratios (10, 50 and 200). The results in Table 5 show that INOLML outperforms other approaches by a large margin. This result is even more remarkable when studying the noise rate of 40%. For CIFAR100, we do not show results with imbalance ratio > 10 since for larger imbalance ratios, it was impossible to build validation sets with 10 samples per class for the minority classes. Nevertheless, for the two CIFAR100 problems, our method shows substantially better

Table 5: Test accuracy (%) of INOLML and other SOTA methods on CIFAR10 and CIFAR100 imbalanced learning mixed with symmetric noise. The reported results are from (Zhang & Pfister, 2021) and (Wei et al., 2021).

| Dataset | CIFAR10 | | | | | | CIFAR100 | |
|---------|---------|-----|-----|-----|-----|-----|----------|-----|
| Noise ratio | 0.2 | | | 0.4 | | | 0.2 | 0.4 |
| Imbalance ratio | 10 | 50 | 200 | 10 | 50 | 200 | 10 | 10 |
| CRUST | 65.7 | 41.5 | 34.3 | 59.5 | 32.4 | 28.8 | - | - |
| LDAM | 82.4 | - | - | 71.4 | - | - | 48.1 | 36.7 |
| LDAM-DRW | 83.7 | - | - | 74.9 | - | - | 50.4 | 39.4 |
| BBN | 80.4 | - | - | 70.0 | - | - | 47.9 | 35.2 |
| HAR-DRW | 82.4 | - | - | 77.4 | - | - | 46.2 | 37.4 |
| ROLT-DRW | 85.5 | - | - | 82.0 | - | - | 52.4 | 46.3 |
| FSR | 85.7 | 77.4 | 65.5 | 81.6 | 69.8 | 49.5 | - | - |
| **INOLML** | **90.1** | **80.1** | **66.6** | **89.1** | **78.1** | **61.6** | **59.8** | **56.1** |

Table 6: Test accuracy (%) of INOLML and previous methods in open-set noise using WideResnet28-10 with 10 samples per class for validation.

| Method | ImageNet | CIFAR100 | BOTH |
|--------|----------|----------|------|
| RoG (Patrini et al., 2017) | 83.4 | 87.1 | 84.4 |
| L2R (Ren et al., 2018) | 81.8 | 81.8 | 85.0 |
| Distill (Zhang et al., 2020) | 94.0 | 92.3 | 93.0 |
| **INOLML** | **94.5 ± 0.1** | **93.6 ± 0.0** | **93.6 ± 0.1** |

results than previous SOTA methods. Our method can, therefore, be considered the new SOTA for this imbalanced noisy-label learning benchmark with Resnet32.

### 5.7 Open-set noise

Open-set noise occurs when training images may belong to classes outside the $C$ classes in $\mathcal{D}$. We consider 3 benchmarks using CIFAR10 contaminated with images from CIFAR100 and ImageNet (Lee et al., 2019b). Table 6 shows results from INOLML, Distill (Zhang et al., 2020) and other meta-learning methods (Patrini et al., 2017; Ren et al., 2018; Zhang & Pfister, 2021), where INOLML achieves the best performance in all cases. Comparing to Distill, INOLML is 0.5% to 1% better, despite the selected validation set $\mathcal{D}^{(v)}$ being largely contaminated with noisy samples (up to 40%), as shown in Figure 3b. However, such performance on open noise contrasts with our observation in the 80% symmetric noise settings, where just 30% noise rate in $\mathcal{D}^{(v)}$ degrades the performance of INOLML, compared to Distill model (Table 1). Such difference might be attributed to the out-of-distribution characteristic of open-set noise. As open-set noisy-label datasets contain samples that do not belong to the set of known classes, such samples might help regularise the learning on mislabelled data, reducing the effect of confirmation bias, resulting in a better performance.

### 5.8 Real-world datasets

Tables 7 and 8 show the results of INOLML and other SOTA approaches on real-world datasets. Table 7 shows the performance on mini-WebVision with 2 popular models: Resnet50 (RN50) and InceptionResnetV2 (InRN), while Table 8 shows results on four different noise ratios evaluated on Red mini-ImageNet with 32 × 32 images with 2 popular models: Resnet18 (RN18) and InceptionResnetV2 (InRN). INOLML is the new SOTA on mini-WebVision with Resnet50 model and Red mini-ImageNet. We note that our method is more efficient in terms of memory footprint than most of Co-training based approaches (Li et al., 2020; Cordeiro et al., 2021; Xu et al., 2021) evaluated on Red Mini-ImageNet since we use only a single PreAct Resnet18

Table 7: Prediction accuracy (%) on the real-world dataset mini-WebVision with Resnet50, evaluated on Webvision and ImageNet test sets; the results of other methods are from (Zhang & Pfister, 2021; Cordeiro et al., 2021) or from their original papers.

| Method | Base | Num. Params | WebVision | ImageNet |
|---|---|---|---|---|
| HAR | InRN | 56M | 75.5 | 57.4 |
| GJS | RN50 | 23M | 78.0 | 74.4 |
| MW-Net | InRN | 56M | 74.5 | 72.6 |
| UNICON | 2×InRN | 112M | 77.6 | 75.3 |
| MOIT | RN18 | 11M | 78.8 | - |
| SSR | InRN | 56M | 80.9 | 75.8 |
| C2D | 2× RN50 | 46M | 79.4 | **78.6** |
| Bayesian DivideMix++ | 2× RN50 | 23M | 80.12 | 78.51 |
| NCR | RN50 | 23M | 80.5 | - |
| BtR | InRN | 56M | 80.9 | 76.0 |
| CC | InRN | 56M | 79.4 | 76.1 |
| TLC | RN50 | 23M | 79.1 | 75.4 |
| FSR | RN50 | 23M | 74.9 | 72.3 |
| **INOLML** | RN50 | 23M | **81.7** | 78.1 |

Table 8: Prediction accuracy (%) on the real-world dataset Red mini-ImageNet dataset. The results of other methods are from (Zhang & Pfister, 2021; Cordeiro et al., 2021) or from their original papers.

| Method | Backbone | № parameters (millions) | Noise ratio 0.2 | 0.4 | 0.6 | 0.8 |
|---|---|---|---|---|---|---|
| CE | 2× RN18 | 22 | 47.36 | 42.70 | 37.30 | 29.76 |
| Mix Up | 2× RN18 | 22 | 49.10 | 46.40 | 40.58 | 33.58 |
| DivideMix | 2× RN18 | 22 | 50.96 | 46.72 | 43.14 | 34.50 |
| MentorMix | InRN | 56 | 51.02 | 47.14 | 43.80 | 33.46 |
| PropMix | 2× RN18 | 22 | 61.24 | 56.22 | 52.84 | 43.42 |
| FaMUS | 2× RN18 | 22 | 51.42 | 48.06 | 45.10 | 35.50 |
| InstanceGM | 2× RN18 | 22 | 58.38 | 52.24 | 47.96 | 39.62 |
| InstanceGM-SS | 2× RN18 | 22 | 60.89 | 48.06 | 45.10 | 35.50 |
| **INOLML** | RN18 | 11 | **63.23** | **58.21** | **53.39** | **45.32** |

with meta-learning instead of two separate PreAct Resnet18. On mini-WebVision benchmark, our method can also achieve significantly better performance despite using a smaller architecture in terms of the number of parameters compared to previous methods.

# 6 Ablation studies

This section studies the factors affecting the optimization in (9). In the lower-lever optimization of (9), the function Info(.) selects samples that maximise the training sample weight $\omega$ from (10), as well as samples that maximise the maximum "information content". We, therefore, carry out an ablation study about the importance of this factor by replacing Info(.) in (9) with the MaxWeight(.) in Eq. (11) and show the results in Table 9. We also study the role of the Clean(.) utility function in (9) by optimising only the lower-level of (9) (see Info(.) **only**). This ablation study is conducted on CIFAR10 and CIFAR100 under 40% asymmetric noise and 20% symmetric noise with imbalanced data. Overall, each component improves model performance compared to naively optimising the average weight in (10). Naively selecting samples based on (10) facilitates the overfiting of the noisy-label samples, leading to confirmation bias. The framework

Table 9: Test accuracy (%) on CIFAR10 and CIFAR100 under asymmetric and imbalanced noisy-label problems, where IR denotes the imbalance ratio. The $1^{st}$ row shows the results of the optimization of the average of weight (col. **Average Weight in** (10)) instead of (9). The $2^{nd}$ row shows the results of optimising the lower part of (9) (col. Info(.) **Only**) without the upper part of (9) Clean(.). The last row (**Whole** (9)) shows our final model result.

| Replace Info with MaxWeight | Info(.) Only | Equation (9) | Asymmetric | | Symmetric | | |
|---|---|---|---|---|---|---|---|
| | | | CIFAR10 | | CIFAR10 | | |
| | | | WRN IR=1 | RN29 IR=1 | RN32 IR=10 | RN32 IR=50 | RN32 IR=200 |
| ✓ | | | 91.0 | 56.6 | 68.8 | 37.6 | 23.4 |
| | ✓ | | 92.1 | 89.3 | 89.0 | 79.1 | 65.9 |
| | | ✓ | 94.1 | 89.8 | 90.1 | 80.1 | 66.6 |

mitigates this problem by using Clean(.) limiting the noise in the clean validation set, while Info(.) prevents the gradient to go toward a single wrong direction.

As shown in Table 9, the impact of Clean(.) varies across different settings. Consequently, applying a uniform optimization framework for all noise types, noise rate and imbalanced ratio may result in unnecessary overhead and potentially suboptimal performance. A promising future direction could involve designing an adaptive framework that automatically determines the optimal training strategy by estimating the class imbalance and noise rate, thereby minimizing framework overhead and optimizing performance.

To demonstrate the effectiveness of INOLML, we measure the distributions of the sample weights $\omega$ during the training process compared to other meta reweighting methods, as demonstrated in Figure 4. Recall that INOLML targets the maximisation of $\omega$ for training samples of high utility, so it is important to measure how $\omega$ progresses during training. The evaluation is conducted on CIFAR-100 with 80% symmetric noise setting, and we compare against FSR (Zhang & Pfister, 2021) and Distill (Zhang et al., 2020), as they are the most closely related frameworks to our approach. The results in Figure 4 show that overall, our method provides higher weight for clean samples and lower weight for noisy samples, compared to other meta reweighting methods such as FSR (Zhang & Pfister, 2021) or Distill (Zhang et al., 2020), demonstrating the effectiveness of our method.

We also compare the validation set produced from (9) with sets built with random sampling and most confident sampling based on the highest confidence scores. Figure 5 shows that the most confident sampling has inferior results compared to random sampling. Nevertheless, INOLML with built-in high utility validation set formation shows the best results.

Traditional meta-learning approaches (Dehghani et al., 2017b; Shu et al., 2019) always keep the clean validation set separate from the training set, while our method iteratively extracts $D^{(v)}$ from the training set. It can be argued that this non-separation of the training and validation sets can cause confirmation bias to happen during training. Hence, we evaluate our approach in a scenario where the candidate samples to form the validation set is always separate from the training set during training. However, our results show that such separate validation set causes a 2% drop in accuracy, on average. This can be explained by the smaller size of the training set and the restriction in potential choices for validation samples.

A final discussion point is the time needed for the INOLML training. Using CIFAR10 with uniform noise, the Distill model takes around 5 and 29 hours to train the Resnet29 and WideResnet28-10 models, respectively. When integrating our method with Distill, training slightly increases to 5.5 hours on Resnet29 and 31 hours on WideResnet28-10. Hence, our algorithm adds around 10% to the training time for optimising the validation set, which happens once per epoch. The experiment above was conducted on a single NVIDIA V100 GPU. We also compare the training time of our approach with other recently proposed methods using the PreAct Resnet18 model on CIFAR100 dataset with 40% uniform noise using a single V100 GPU. While our approach takes 10.5h, DivideMix (Li et al., 2020) takes 8.25 hours, CausalNL(Yao et al., 2021) takes around 12.5 hours,

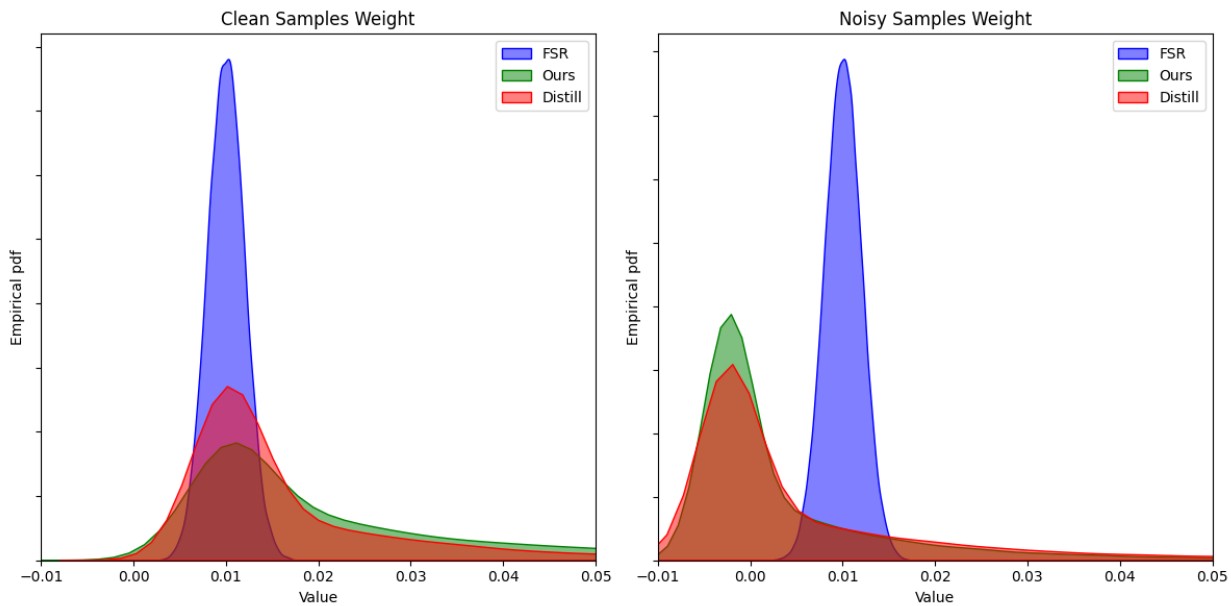

Figure 4: Weight distribution of samples from different data reweighting methods, under the setting CIFAR100 with 0.8 uniform noise ratio.

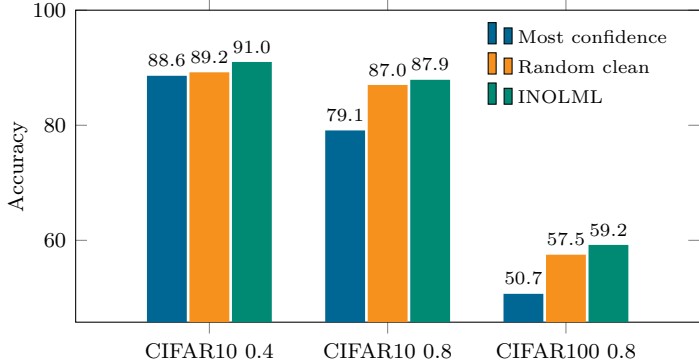

Figure 5: Accuracy (%) of our INOLML using different sample selection methods under uniform label noises.

and MOIT (Ortego et al., 2021b) takes 8 hours, indicating that our approach has competitive training time compared to recently proposed methods.

## 7 Conclusion

We presented a novel meta-learning approach, called INOLML, that automatically and progressively selects a pseudo-clean validation set from a noisily-labelled training set. This selection is based on our proposed validation set utility criteria that take into account sample informativeness, class-balanced distribution, and label cleanliness. Our proposed method is more effective and practical than prior meta-learning approaches since we do not require manually-labelled samples to be included in the validation set. Compared with other meta-learning approaches that do not require a manually labelled validation set (e.g. FSR or FaMUS), INOLML has demonstrated to be more robust to high noise rate problems and able to achieve SOTA results on several synthetic and real-world label noise benchmarks. In fact, INOLML has SOTA results in mini-WebVision, Red mini-ImageNet, open-set noise, long-tailed + symmetric noise for CIFAR-10/-100, symmetric and asymmetric CIFAR-10/-100, and imbalanced learning, with substantial improvements (e.g., CIFAR-10 80% symmetric, CIFAR-100 40% symmetric, imbalanced and noisy-label imbalanced benchmarks, and mini-WebVision).

A limitation of our approach is that the model can suffer from confirmation bias as it is based on a single network. As future work, we will tackle this problem by incorporating co-teaching in our meta-learning algorithm. Another limitation is the greedy and complex bi-level optimization to form the validation set in (9), which can be improved in two ways: 1) the complexity can be reduced by replacing the bi-level optimization with a single-level optimization, and 2) the greedy strategy can be replaced by a relaxation method to solve the combinatorial optimization problem. Additionally, optimising the clean validation set once per epoch is not ideal since the validation set can be outdated by the end of epoch. This issue will be addressed by updating the clean validation set more regularly. Finally, another point missing from this paper is a theoretical analysis of the proposed criteria to characterise the utility of the meta-learning validation set. In particular, we plan to study why the relaxation of the assumption of clean and balanced validation set made by Ren et al. (2018) in (10) still works for our validation set that has pseudo-clean, balanced and informative samples.

### Acknowledgments

G.C. and C.N. are supported by the Engineering and Physical Sciences Research Council (EPSRC) through grant EP/Y018036/1.

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

## A    Real world noise results

Table 10: Prediction accuracy (%) on real-world datasets. *(left):* WebVision with Resnet50, evaluated on Webvision and ImageNet test sets; and *(right):* Red Mini-ImageNet. The results of other methods are from (Zhang & Pfister, 2021; Cordeiro et al., 2021) or from original papers.

| Method | WebVision | | ImageNet | |
|---|---|---|---|---|
| | top-1 | top-5 | top-1 | top-5 |
| HAR | 75.5 | 90.7 | 57.4 | 82.4 |
| Co-teaching | 63.6 | 85.2 | 61.5 | 84.7 |
| Iterative-CV | 65.2 | 85.3 | 61.6 | 85.0 |
| MentorNet | 63.0 | 81.4 | 63.8 | 85.8 |
| CRUST | 72.4 | 89.6 | 67.4 | 87.8 |
| GJS | 78.0 | 90.6 | 74.4 | 91.2 |
| MW-Net | 74.5 | 88.9 | 72.6 | 88.8 |
| UNICON | 77.6 | 93.4 | 75.3 | **93.7** |
| MOIT | 78.8 | - | - | - |
| FSR | 74.9 | 88.2 | 72.3 | 87.2 |
| **INOLML** | **81.7** | **93.8** | **78.1** | 92.9 |

| Method | Noise ratio | | | |
|---|---|---|---|---|
| | 0.2 | 0.4 | 0.6 | 0.8 |
| CE | 47.36 | 42.70 | 37.30 | 29.76 |
| Mix Up | 49.10 | 46.40 | 40.58 | 33.58 |
| DivideMix | 50.96 | 46.72 | 43.14 | 34.50 |
| MentorMix | 51.02 | 47.14 | 43.80 | 33.46 |
| PropMix | 61.24 | 56.22 | 52.84 | 43.42 |
| FaMUS | 51.42 | 48.06 | 45.10 | 35.50 |
| **INOLML** | **63.23** | **58.21** | **53.39** | **45.32** |

## B    Implementation Details

All CIFAR experiments use batches of size 100, which are trained on a single GPU. Similar to the Distill noise model (Zhang et al., 2020), we use $p = 5, k = 20$ for CIFAR experiments, except the ones with the imbalance setting.

For Red Mini-ImageNet experiments, we trained the model on a single GPU with batches of size 100, with $p = 5, k = 10$.

For the WebVision experiment, we use $p = 4, k = 8$ with 4 NVIDIA V100 GPU and batches of size 16. All experiments use $N = 200, K = 50, \kappa = 0.9$.

In practice, to reduce the computational cost of the optimisation in (9), we replace the pseudo-clean set $\mathcal{D}^{(c)}$ with the following subset:

$$\widetilde{\mathcal{D}}^{(c)} = \left\{ (\mathbf{x}_i, \mathbf{y}_i) : (\mathbf{x}_i, \mathbf{y}_i) \in \mathcal{D}^{(c)} \wedge \underset{k \in \{1, \ldots, C\}}{\operatorname{argmax}} \mathbf{y}_i(k) = \underset{k \in \{1, \ldots, C\}}{\operatorname{argmax}} \tilde{\mathbf{y}}_i(k) \right\}.$$

In summary, we aim to progressively refine the pseudo clean set $\mathcal{D}^{(c)}$, making it cleaner over time. $\widetilde{\mathcal{D}}^{(c)}$ is a subset of $\mathcal{D}^{(c)}$, containing $N$ randomly-selected samples $(\mathbf{x}_i, \mathbf{y}_i)$ of each class in $\mathcal{D}^{(c)}$ that have their observed labels $\mathbf{y}_i$ consistent with the corresponding moving average robust label computed with the average prediction over the last $E$ epochs, as in:

$$\tilde{\mathbf{y}}_i = \kappa \tilde{\mathbf{y}}_i + (1 - \kappa) \frac{1}{E} \sum_{e=1}^{E} f_\theta(\mathbf{x}_i),$$

with $\kappa \in [0, 1]$ being a hyper-parameter.

## C    Additional Results of Symmetric Noise on CIFAR Datasets

We provide additional symmetric noise results of our proposed method and the Distill model (Zhang et al., 2020) in Table 11. Note that our method is markedly better than Distill, particularly for the simpler model

(RN29) with few samples per class (1 and 5) in the validation set. For the more complex model (WRN) and large validation set (10 samples per class), our method is still better than Distill, except for CIFAR100 at 0.8 symmetric noise rate.

As shown in Table 11, our method achieves only moderate improvements over Distill when the validation set size is large (10 samples per class). However, it demonstrates significant improvements when the validation set size is smaller, or the noise rate is higher (e.g., 11.5% improvement for CIFAR-100 with 0.8 uniform noise and 4% improvement for CIFAR-100 with 0.2 uniform noise, using 5 samples per class in the validation set). Similarly, our method demonstrates significant improvements on imbalanced and noisy benchmarks, as shown in Table 5. This property highlights that our method performs better in scenarios with an extreme scarcity of clean data, especially when the dataset is both noisy and imbalanced. A possible explanation for this phenomenon is that when the number of clean training samples is limited, the informativeness of each sample becomes more critical for training compared to scenarios with abundant clean training data.

Table 11: Test accuracy (in %) comparison between our method ("INOLML") and the Distill noise ("DN") on symmetric noise using 1, 5 and 10 samples per class in the validation set on two backbone models: Resnet29 ("RN29") and Wideresnet28-10 ("WRN"). The results of the Distill model with WideResnet28-10 are collected from (Zhang et al., 2020). Recall that the Distill needs a clean set, while INOLML works with a pseudo-clean set.

| Method | Val. Set size | Dataset | | | | | |
| | | CIFAR10 | | | CIFAR100 | | |
| | | 0.2 | 0.4 | 0.8 | 0.2 | 0.4 | 0.8 |
|---|---|---|---|---|---|---|---|
| DN-RN29 | 1 | $87.0 \pm 0.5$ | $85.3 \pm 0.5$ | FAIL | $58.9 \pm 0.5$ | $55.8 \pm 0.7$ | FAIL |
| INOLML-RN29 | | $90.3 \pm 0.2$ | $89.1 \pm 0.5$ | $79.1 \pm 0.3$ | $65.9 \pm 0.2$ | $61.5 \pm 0.4$ | $55.1 \pm 0.6$ |
| DN-RN29 | 5 | $90.7 \pm 0.3$ | $89.0 \pm 0.3$ | $83.5 \pm 0.2$ | $62.6 \pm 0.4$ | $58.8 \pm 0.5$ | $48.5 \pm 0.5$ |
| INOLML-RN29 | | $90.9 \pm 0.2$ | $90.9 \pm 0.1$ | $87.4 \pm 0.2$ | $66.6 \pm 0.1$ | $65.7 \pm 0.1$ | $59.0 \pm 0.5$ |
| DN-RN29 | 10 | $91.0 \pm 0.2$ | $89.2 \pm 0.1$ | $87.0 \pm 0.1$ | $63.7 \pm 0.2$ | $60.5 \pm 0.2$ | $57.5 \pm 0.5$ |
| INOLML-RN29 | | $92.2 \pm 0.1$ | $91.0 \pm 0.1$ | $87.9 \pm 0.2$ | $67.1 \pm 0.1$ | $66.3 \pm 0.1$ | $59.2 \pm 0.2$ |
| DN-WRN | 1 | $95.4 \pm 0.6$ | $94.5 \pm 1.0$ | $87.9 \pm 5.1$ | $77.4 \pm 0.4$ | $75.5 \pm 1.1$ | $62.1 \pm 1.2$ |
| INOLML-WRN | | $96.0 \pm 0.2$ | $95.9 \pm 0.2$ | $94.3 \pm 0.2$ | $81.6 \pm 0.2$ | $79.5 \pm 0.2$ | $73.6 \pm 0.3$ |
| DN-WRN | 5 | $96.4 \pm 0.0$ | $95.5 \pm 0.6$ | $91.8 \pm 3.0$ | $80.4 \pm 0.5$ | $79.6 \pm 0.3$ | $73.6 \pm 1.5$ |
| INOLML-WRN | | $96.4 \pm 0.1$ | $96.2 \pm 0.1$ | $94.6 \pm 0.2$ | $82.2 \pm 0.2$ | $81.5 \pm 0.2$ | $74.5 \pm 0.2$ |
| DN-WRN | 10 | $96.2 \pm 0.2$ | $95.9 \pm 0.2$ | $93.7 \pm 0.5$ | $81.2 \pm 0.7$ | $80.2 \pm 0.3$ | $\mathbf{75.5 \pm 0.2}$ |
| INOLML-WRN | | $\mathbf{96.9 \pm 0.1}$ | $\mathbf{96.6 \pm 0.1}$ | $\mathbf{95.0 \pm 0.2}$ | $\mathbf{82.0 \pm 0.2}$ | $\mathbf{81.3 \pm 0.2}$ | $74.7 \pm 0.1$ |

# D   Additional Results of Semantic Noise on CIFAR100 Dataset

We provide additional results for semantic noise introduced by RoG (Lee et al., 2019a) comparing our proposed method and the Distill model (Zhang et al., 2020) in Table 12. Both methods utilize the ResNet29 architecture, with validation set sizes ranging from 1, 5, to 10 samples per class.

Table 12: Test accuracy (%) of INOLML on CIFAR100 with the semantic noise introduced by RoG (Lee et al., 2019b), in comparison with Distill using a validation set $\mathcal{D}^{(v)}$ of sizes 1, 5 and 10 samples per class on Resnet29 model. The superscript $^{\mathrm{T}}$ indicates the need for clean validation sets.

| Method | $|\mathcal{D}^{(v)}|$ | Accuracy |
|---|---|---|
| Distill$^{\mathrm{T}}$ | $1 \times C$ | $58.1 \pm 0.1$ |
| **INOLML** | | $61.5 \pm 0.2$ |
| Distill$^{\mathrm{T}}$ | $5 \times C$ | $61.3 \pm 0.3$ |
| **INOLML** | | $63.1 \pm 0.3$ |
| Distill$^{\mathrm{T}}$ | $10 \times C$ | $61.5 \pm 0.3$ |
| **INOLML** | | $\mathbf{63.5 \pm 0.1}$ |

