# OpenReview forum: "Maximising the Utility of Validation Sets for Imbalanced Noisy-label Meta-learning"
_TMLR — Accepted by TMLR_

### Review · Reviewer_YQ3e · 2024-12-28

**Summary Of Contributions:**

This paper proposes a new automatic meta-learning pipeline that involves choosing samples from the validation set using a utility function that balances 3 objectives (sample informativeness, label cleanliness and balanced class distribution). Experiment results show that the proposed method

**Audience:**

Yes

**Claims And Evidence:**

Yes

**Requested Changes:**

The writing of argmax equations (e.g. Eq (9)) is nonstandard. It is unusual to use the same symbol for the left-hand-side as well as the quantify being argmax'd. Something like $x^* = argmax_{x} f(x)$ would make more sense.

Additional experiments on some implementation choices would be good.

**Strengths And Weaknesses:**

Strengths

The method is intuitive and not too complicated.

The experiment is rather extensive (in terms of baselines).

Weaknesses:

The overall approach sometimes looks a little brute-force/naive, in particular the "greedy strategies" mentioned right after Eq(14). This is not an issue by itself, but I'd like to see some experiments by replacing these with simple variants.

Some baselines look a little old and the proposed method is not always better.

I worry that the proposed method encodes some implicit bias about how the noisy dataset is constructed. For example, the cleanliness heuristic may not work if the label is noisy in a different manner, or is similar in a manifold not captured by $z$. I understand that this is, however, very hard to test.

---

> ### Author Response · Authors · 2025-01-09
>
> # **Response to Reviewer YQ3e**
>
> **We greatly appreciate the time and effort the Reviewer dedicated to our paper. Here are our responses to all concerns raised by the Reviewer.**
>
> ## **Weakness**
>
> ### **Q3.1**
> *The overall approach sometimes looks a little brute-force/naive, in particular the "greedy strategies" mentioned right after Eq(14). This is not an issue by itself, but I'd like to see some experiments by replacing these with simple variants.*
>
> **R3.1**:
> We provide additional results when replacing Eq.(14) by simply selecting samples with the minimum loss when refining the candidate set for the validation set (Table 1 below), as clean samples are likely to have small losses. As shown, the performance of the method slightly drops compared to using our Eq.(14).
>
> | **Method**  | **Accuracy** (0.4) | **Accuracy** (0.8) |
> |-------------|--------------------|--------------------|
> | MinLoss     | 90.5              | 87.3              |
> | Ours        | 90.9              | 87.9              |
>
> **Table 1**: Test accuracy (%) on the CIFAR-10 dataset with symmetric noise. **MinLoss** represents the results when the cleanliness objective in Eq.(14) is replaced with selecting samples with the minimum losses.
>
> ---
>
> ### **Q3.2**
> *Some baselines look a little old and the proposed method is not always better.*
>
> **R3.2**:
> We have included several new noisy label learning baselines published in 2024, such as **PSDC** [1] and **Bayesian DivideMix++** [2] (highlighted in the revised paper).
>
> Regarding experiments where our method is not state-of-the-art (e.g., ImageNet or the asymmetric noise benchmark), it is important to note that our primary competitors are other meta-reweighting frameworks, including **Famus** [3], **FSR** [4], **L2R** [5], **MWN** [6], **GDW** [7], and **Distill** [8]. Additionally, methods such as **C2D** [9] and **Bayesian DivideMix++** [2] leverage contrastive learning or involve more parameters than our framework.
>
> ---
>
> ### **Q3.3**
> *I worry that the proposed method encodes some implicit bias about how the noisy dataset is constructed. For example, the cleanliness heuristic may not work if the label is noisy in a different manner, or is similar in a manifold not captured by z. I understand that this is, however, very hard to test.*
>
> **R3.3**:
> We have demonstrated the effectiveness of our method across diverse benchmarks with various noise types, including open-set noise, uniform noise, asymmetric noise, and real-world datasets like WebVision and Red mini-ImageNet.
>
> To further address this concern, we present additional results on semantic noise introduced by **RoG** [10]. This type of noise is generated by a neural network trained on 20% of the CIFAR100 dataset. Since this noise is produced by a neural network, noisy samples often have embeddings \\( z \\) similar to samples from incorrectly annotated classes. The results are shown in Table 2.
>
>
>
> | Method              | \\(\|\\mathcal{D}^{(v)}\|\\) | **Accuracy**      |
> |---------------------|-----------------------------|-------------------|
> | Distill\\(\^{T}\\)  | \\(1 \\times C\\)          | 58.1 ± 0.1          |
> | **INOLML**             |                         | 61.5 ± 0.2          |
> | Distill\\(\^{T}\\)    | \\(5 \\times C\\)          | 61.3 ± 0.3          |
> | **INOLML**             |                         | 63.1 ± 0.3          |
> | Distill\\(\^{T}\\)    | \\(10 \\times C\\)         | 61.5 ± 0.3          |
> | **INOLML**             |                         | **63.5 ± 0.1**      |
>
>
>
>
>
> **Table 2**: Test accuracy (%) of INOLML on CIFAR100 with semantic noise introduced by RoG [10]. Results are compared to **Distill** using validation sets \\( \\mathcal{D}^{(v)} \\) of sizes 1, 5, and 10 samples per class on the ResNet29 model. Superscript \\( T \\) indicates the use of clean validation sets.
>
> ---
>
> ## **Request Changes**
>
> ### **Change3.1**
> *The writing of argmax equations (e.g. Eq (9)) is nonstandard. It is unusual to use the same symbol for the left-hand-side as well as the quantify being argmax'd. Something like \\( x\^\* = \\arg\\max_x f(x) \\) would make more sense.*
>
> **Implementation3.1**:
> We sincerely thank the Reviewer for the suggestion. We have updated Eq.(9) based on the provided feedback.
>
> ---
>
> ### **Change3.2**
> *Additional experiments on some implementation choices would be good.*
>
> **Implementation3.2**:
> Please refer to the additional results provided for **Q3.1** and **Q3.3** in Tables 1 and 2 above.
>
> ---

---

> > ### Author Response · Authors · 2025-01-09
> >
> > # **References**
> >
> > [1] Sihan Bai. Pairwise similarity distribution clustering for noisy label learn-
> > ing. ArXiv, abs/2404.01853, 2024.
> >
> >
> > [2] Bhalaji Nagarajan, Ricardo Marques, Eduardo Aguilar, and Petia Radeva.
> > Bayesian dividemix++ for enhanced learning with noisy labels. Neural
> > Networks, 172:106122, 2024.
> >
> >
> > [3] Youjiang Xu, Linchao Zhu, Lu Jiang, and Yi Yang. Faster meta update
> > strategy for noise-robust deep learning. In CVPR, 2021.
> >
> >
> > [4] Zizhao Zhang and Tomas Pfister. Learning fast sample re-weighting with-
> > out reward data. In ICCV, 2021.
> >
> >
> > [5] Mengye Ren, Wenyuan Zeng, Binh Yang, and Raquel Urtasun. Learning
> > to reweight examples for robust deep learning. In ICML, 2018.
> >
> >
> > [6] Jun Shu, Qi Xie, Lixuan Yi, Qian Zhao, Sanping Zhou, Zongben Xu, and
> > Deyu Meng. Meta-weight-net: Learning an explicit mapping for sample
> > weighting. In NeurIPS 2019.
> >
> >
> > [7] Can Chen, Shuhao Zheng, Xi Chen, Erqun Dong, Xue Liu, Hao Liu, and
> > Dejing Dou. Generalized data weighting via class-level gradient manipula-
> > tion. In NeurIPS, 2021.
> >
> >
> > [8] Zizhao Zhang, Han Zhang, Sercan ¨O. Arik, Honglak Lee, and Tomas Pfister.
> > Distilling effective supervision from severe label noise. In CVPR, 2019.
> >
> >
> > [9] Evgenii Zheltonozhskii, Chaim Baskin, Avi Mendelson, Alexander M. Bron-
> > stein, and Or Litany. Contrast to divide: Self-supervised pre-training for
> > learning with noisy labels. In WACV, 2022.
> >
> >
> > [10] Kimin Lee, Sukmin Yun, Kibok Lee, Honglak Lee, Bo Li, and Jinwoo Shin.
> > Robust inference via generative classifiers for handling noisy labels. In
> > ICML, 2019.

---

### Review · Reviewer_Mm2x · 2024-12-29

**Summary Of Contributions:**

In this paper, a meta-learning (ML) method for training classifiers using noisy and class-imbalanced labels. The previous method ``Distill'' proposed by Zhang et al. has provided an ML framework that meta-learns both sample importance weights and NN-based label correction for noisy samples. However, this method relies on a clean manually crafted validation set, which is costly in large-scale training.

To address this issue, this study introduces a novel method to automatically construct a high-quality validation set based on three criteria: (i) sample informativeness, (ii) class-balanced distribution, and (iii) label cleanliness. Notably, the informativeness and label cleanliness criteria are original contributions introduced by the authors. The performance of the proposed method is comprehensively evaluated using standard experimental setups similar to existing research. The results demonstrate that the proposed method outperforms previous methods in most cases. Additionally, section 6 presents experimental evidence supporting the validity of the proposed method's key ideas.

**Audience:**

Yes

**Broader Impact Concerns:**

No broader concerns.

**Claims And Evidence:**

No

**Requested Changes:**

* [Section 3.3] Please extend the explanation of _Distill_. Although the key equations are shown in the text, the idea behind each content is not clearly explained. What is the idea behind introducing $\lambda$ and pseudo-labels? Of course, consulting Zhang et al. solves this, however, those should be explained in this paper as the proposed method heavily relies on the ``Distill'' framework.
 - Also, ** where $\lambda_i=0.9$** below equation (5) is weird as it should be optimized.
* [Section 5-6] The proposed method seems to be mostly effective when the class-imbalance ratio is large, while the performance improvement over ``Distill'' is moderate in class-balanced cases. Please add an explanation on this point. Why the proposed method is so effective in class-imbalanced cases (or equivalently the previous methods are ineffective)? Could you conduct some ablation studies that directly validate the mechanism by which the proposed method operates in label-imbalanced cases?

**Strengths And Weaknesses:**

## Strength
* The proposed method for automatically curating high-quality validation data for meta-learning is relevant to TMLR readerships, and the proposed criteria are, at least to me, interesting.
* The results demonstrate that SoTA performance is achieved in many situations.

## Weakness
* [Section3.3] The explanation of the existing method,  which the proposed method relies on, is too concise, making it difficult to understand the ideas behind each component.
* [section6] The validation of the proposed method's idea is somewhat insufficient. In particular, while the proposed method significantly outperforms other methods when the class-imbalance ratio is high, the mechanism behind this advantage has not been adequately examined.

---

> ### Author Response · Authors · 2025-01-09
>
> # **Response to Reviewer Mm2x**
>
> **We greatly appreciate the time and effort the Reviewer dedicated to our paper. Here are our responses to all concerns raised by the Reviewer.**
>
> ## **Weakness**
>
> **Q2.1**: *The explanation of the existing method, which the proposed method relies on, is too concise, making it difficult to understand the ideas behind each component.*
>
> **R2.1**:
> We sincerely thank the Reviewer for the suggestion. Section 3.3 provides details about the framework proposed by Zhang et al. [1], which is the baseline of our method. We have included additional details (highlighted in red) about their framework and the rationale behind its components in Section 3.3 in our revision.
>
> **Q2.2**: *The validation of the proposed method's idea is somewhat insufficient. In particular, while the proposed method significantly outperforms other methods when the class-imbalance ratio is high, the mechanism behind this advantage has not been adequately examined.*
>
> **R2.2**:
> As shown in Table 11 in the Appendix of our paper, our method achieves only moderate improvements over Distill when the validation set size is large (10 samples per class). However, it demonstrates significant improvements when the validation set size is smaller or the noise ratio is higher (e.g., 11.5% improvement for CIFAR-100 with 0.8 uniform noise and 4% improvement for CIFAR-100 with 0.2 uniform noise, using 5 samples per class in the validation set). This property highlights that our method performs better in scenarios with an extreme scarcity of clean data, especially when the dataset is both noisy and imbalanced.
>
> In contrast to existing meta-reweighting approaches for noisy and imbalanced data, such as Zhang et al. [2] and Xu et al. [3], our method focuses directly on maximizing the informativeness of the validation samples, making it particularly effective in label-imbalanced settings.
>
> ---
>
> ## **Request Changes**
>
> **Change2.1**:
> *[Section 3.3] Please extend the explanation of Distill. Although the key equations are shown in the text, the idea behind each content is not clearly explained. What is the idea behind introducing pseudo-labels? Of course, consulting Zhang et al. solves this, however, those should be explained in this paper as the proposed method heavily relies on the "Distill" framework. Also, "where \\( \\lambda = 0.9 \\)" below equation (5) is weird as it should be optimized.*
>
> **Implementation2.1**:
> Common approaches for learning with noisy labels include upweighting clean samples while downweighting noisy samples during training or relabeling noisy samples and uniformly minimizing their losses. The Distill framework proposed by Zhang et al. [1] incorporates both of these strategies. In their framework, each training sample is associated with two labels, \\( \\hat\{\\mathbf\{y\}\}\_i(\\lambda_i) \\) and \\( \\mathbf{y}\_i\^\{\*\}(\\lambda^*_i) \\). For label \\( \\hat{\\mathbf{y}}_i(\\lambda_i) \\), instead of optimizing the parameter \\( \\lambda\_i \\), its value is fixed at 0.9 (close to 1) to ensure that label \\( \\hat{\\mathbf{y}}\_i(\\lambda_i) \\) retains information from the original label. Supervised learning with label \\( \\hat{\\mathbf{y}}_i(\\lambda_i) \\) and the optimized weight \\( \\omega_i \\) effectively corresponds to upweighting clean samples and downweighting noisy samples. On the other hand, the parameter \\( \\lambda^*_i \\) for the new label \\( \\mathbf{y}\_i\^\{\*\}(\\lambda\^\*\_i) \\) is optimized with Eq. 6 to infer the correct label for the samples, enabling uniform supervised learning.
>
> We have also included an explanation, highlighted in red, in Section 3.3 of the revised paper.
>
> ---
>
> **Change2.2**:
> *[Section 5-6] The proposed method seems to be mostly effective when the class-imbalance ratio is large, while the performance improvement over "Distill" is moderate in class-balanced cases. Please add an explanation on this point. Why is the proposed method so effective in class-imbalanced cases (or equivalently why are the previous methods ineffective)? Could you conduct some ablation studies that directly validate the mechanism by which the proposed method operates in label-imbalanced cases?*
>
> **Implementation2.2**:
> Please refer to our response to the Reviewer’s concern in **R2.2** above. We have also included an explanation (highlighted in red) in Section C of the Appendix in the revised paper. We have conducted ablation studies for our framework under uniform noise with varying imbalance ratios (10, 50, 200), as presented in Table 9 of the paper.
>
>
>
> # **References**
>
>
> [1] Zizhao Zhang, Han Zhang, Sercan ¨O. Arik, Honglak Lee, and Tomas Pfister.
> Distilling effective supervision from severe label noise. In CVPR, 20219.
>
>
> [2] Zizhao Zhang and Tomas Pfister. Learning fast sample re-weighting with-
> out reward data. In ICCV, 2021.
>
>
> [3] Youjiang Xu, Linchao Zhu, Lu Jiang, and Yi Yang. Faster meta update
> strategy for noise-robust deep learning. In CVPR, 2021.

---

### Review · Reviewer_n416 · 2024-12-30

**Summary Of Contributions:**

This paper presents a novel meta-learning approach, INOLML, designed for validation set selection in the context of imbalanced and noisy-label datasets. INOLML is guided by three core principles: sample informativeness, class balance, and label cleanliness. Extensive experimental results on noisy-label and class-imbalanced benchmarks highlight the effectiveness of the proposed method.

**Audience:**

Yes

**Claims And Evidence:**

Yes

**Requested Changes:**

See weakness

**Strengths And Weaknesses:**

(S1) The experiment results are impressive.

(S2) The approach is simple, intuitive, and well-explained, with some design choices supported by ablation studies.

(S3) The paper is generally easy to follow.

(W1) Not all three utility criteria (informativeness, class balance, and label cleanliness) appear strictly necessary for all scenarios. For example, incorporating the label cleanliness objective may not be essential in imbalanced settings without noisy labels, potentially leading to unnecessary computational overhead. In practical applications, it may not be clear whether the dataset is imbalanced, noisy, or both. How do the authors propose handling such situations?

(W2) There is a consistent use of "optimisation" throughout the paper, which appears to be a typographical preference.

---

> ### Author Response · Authors · 2025-01-09
>
> # **Response to Reviewer n416**
>
> **We greatly appreciate the time and effort the Reviewer dedicated to our paper. Here are our responses to all concerns raised by the Reviewer.**
>
> ## **Weakness**
>
> **Q1.1**: *Not all three utility criteria (informativeness, class balance, and label cleanliness) appear strictly necessary for all scenarios. For example, incorporating the label cleanliness objective may not be essential in imbalanced settings without noisy labels, potentially leading to unnecessary computational overhead. In practical applications, it may not be clear whether the dataset is imbalanced, noisy, or both. How do the authors propose handling such situations?*
>
> **R1.1**:
> We sincerely thank the Reviewer for the suggestion. The cleanliness criterion is primarily effective in extreme scenarios, such as when the noise rate is high or the class imbalance ratio is significant. Our method includes a step that estimates pseudo-clean and pseudo-noisy sets, which allows us to approximately infer the imbalance and noise rate of the training dataset. A potential future direction could involve addressing the point raised by the reviewer by developing an adaptive framework that automatically identifies the optimal training scheme for the model after estimating the imbalance and noise rates, thereby reducing overhead and optimizing performance. However, this direction is beyond the scope of the current paper. We have added a brief discussion (highlighted in red text) in the Ablation section of the revised paper.
>
> **Q1.2**: *There is a consistent use of "optimisation" throughout the paper, which appears to be a typographical preference.*
>
> **R1.2**:
> We sincerely thank the Reviewer for noticing this problem. We will update "optimisation" to "optimization" to maintain consistency with American English spelling.

---

### Decision · Action_Editor_Tnrz · 2025-03-04

**Recommendation:** Accept as is

**Comment:**

Initially, the reviewers expressed concerns about two main aspects of the submission. After the rebuttal, the AE confirmed that the paper now meets expectations. The major concerns of the reviewers have been addressed.  Specifically,

- *Explanation of Baseline Components.*
The reviewers noted that the description of the baseline framework was too concise. For instance, Reviewer Mm2x pointed out that while the paper included key equations, the underlying rationale such as the role of pseudo-labels and the mechanism for differentiating between the original and corrected labels was not adequately explained. Reviewer n416 similarly suggested that extending the discussion on how pseudo-labels contribute to handling noisy data would help readers understand the baseline method's strengths and limitations. In response, the authors have substantially expanded Section 3.3, providing a detailed breakdown of the Distill framework, and clarifying why each component (including the fixed label for supervision and the optimized parameter for label correction) is crucial for achieving robust performance.

- *Optimization Strategy for Validation Set Selection.*
The second area of concern was the use of a greedy optimization strategy for selecting validation samples. Both Reviewer YQ3e and Reviewer Mm2x expressed that the “greedy” approach, particularly as presented immediately after Equation (14), appeared somewhat naïve and warranted a deeper justification. The reviewers recommended including additional experiments such as comparisons with alternative strategies (e.g., a simple min-loss selection) to validate the effectiveness of the proposed optimization mechanism. The authors addressed this by incorporating additional experiments and ablation studies that compare the original greedy approach with other methods.

**Audience:**

Some of TMLR's audience would be interested in this paper. The work targets a challenging and relevant problem, which handles imbalanced and noisy label data in meta-learning.

**Claims And Evidence:**

This paper presents a meta-learning approach that automatically curates validation sets by maximizing a utility function based on three criteria: sample informativeness, balanced class distribution, and label cleanliness. The method is designed to tackle the dual challenges of noisy labels and class imbalance by reducing the dependence on manually curated, clean validation sets. The paper provides extensive experiments across diverse noise settings (uniform, asymmetric, semantic, and open-set noise) and varying class imbalance ratios. Ablation studies further confirm the contribution of each component. Major claims are also well-supported.